# Vocabulary In-Context Learning in Transformers: Benefits of Positional Encoding

## Abstract

Numerous studies have demonstrated that the Transformer architecture possesses the capability for in-context learning (ICL). In scenarios involving function approximation, context can serve as a control parameter for the model, endowing it with the universal approximation property (UAP). In practice, context is represented by tokens from a finite set, referred to as a vocabulary, which is the case considered in this paper, *i.e.,* vocabulary in-context learning (VICL). We demonstrate that VICL in single-layer Transformers, without positional encoding, does not possess the UAP; however, it is possible to achieve the UAP when positional encoding is included. Several sufficient conditions for the positional encoding are provided. Our findings reveal the benefits of positional encoding from an approximation theory perspective in the context of ICL.

## 1 Introduction

Transformers have emerged as a dominant architecture in deep learning over the past few years. Thanks to their remarkable performance in language tasks, they have become the preferred framework in the natural language processing (NLP) field. A major trend in modern NLP is the development and integration of various black-box models, along with the construction of extensive text datasets. In addition, improving model performance in specific tasks through techniques such as in-context learning (ICL) (Dong et al. (2024); Brown et al. (2020)), chain of thought (CoT) (Wei et al. (2022b); Chu et al. (2024)), and retrieval-augmented generation (RAG) (Gao et al. (2024)) has become a significant research focus. While the practical success of these models and techniques is well-documented, the theoretical understanding of why they perform so well remains incomplete.

To explore the capabilities of Transformers in handling ICL tasks, it is essential to examine their approximation power. The Universal Approximation Property (UAP) (Cybenko (1989); Hornik et al. (1989); Hornik (1991); Leshno et al. (1993)) has long been a key topic in the theoretical study of neural networks (NNs), with much of the focus historically on feed-forward neural networks (FNNs). Yun et al. (2020) was the first to investigate the UAP of Transformers, demonstrating that any sequence-to-sequence function could be approximated by a Transformer network with fixed positional encoding. Luo et al. (2022) highlighted that a Transformer with relative positional encoding does not possess the UAP. Meanwhile, Petrov et al. (2024b) explored the role of prompting in Transformers, proving that prompting a pre-trained Transformer can act as a universal functional approximator.

However, one limitation of these studies is that, in practical scenarios, the inputs to language models are derived from a finite set embedded in high-dimensional Euclidean space—commonly referred to as a vocabulary. Whether examining the work on prompts in Petrov et al. (2024b) or the research on ICL in Ahn et al. (2024); Cheng et al. (2024), these studies assume inputs from the entire Euclidean space, which differs significantly from the discrete nature of vocabularies used in real-world applications.

### 1.1 Contributions

Starting with the connection between FNNs and Transformers, we turn to the finite restriction of vocabularies and study the benefits of positional encoding. Leveraging the UAP of FNNs, we explore

the approximation properties of Transformers for ICL tasks in two scenarios: one where the inputs are from the entire Euclidean space, and the other where the inputs are from a finite vocabulary.

1. Without the restriction of a finite vocabulary, we establish a connection between FNNs and Transformers in processing ICL tasks, as demonstrated in Lemma 2. Using this lemma, we show that Transformers can function as universal approximators (Lemma 3), where the context serves as control parameters, while the weights and biases of the Transformer remain fixed.

2. When the vocabulary is finite and positional encoding is not used, we prove that single-layer Transformers cannot achieve the UAP for ICL tasks (Theorem 6). However, when the vocabulary is finite and positional encoding is used, it becomes possible for single-layer Transformers to achieve the UAP (Theorem 8). In particular, for Transformers with ReLU activation functions, the conditions on the positional encoding are discussed (Theorem 9).

## 1.2 RELATED WORKS

**Universal approximation property.** Neural networks (NNs), through multi-layer nonlinear transformations and feature extraction, are capable of learning deep feature representations from raw data. From the early feed-forward neural networks (FNNs) (Rosenblatt (1958)), to later advancements like recurrent neural networks (RNNs) (Waibel et al. (1989); Hochreiter & Schmidhuber (1997)), convolutional neural networks (CNNs) (Waibel et al. (1989); Lecun et al. (1998)), and residual neural networks (ResNets) (He et al. (2016)), remarkable progress has been made. As the application of NNs becomes more widespread, efforts have been directed toward understanding the theoretical foundations behind their effectiveness, particularly through the UAP of NNs. Research on the UAP of NNs generally falls into two categories: the first considers networks with any number of neurons in each layer but a fixed number of layers (Cybenko (1989); Hornik et al. (1989); Hornik (1991); Leshno et al. (1993)), while the second examines networks with an arbitrary number of layers but a finite number of neurons in each layer (Lu et al. (2017); Park et al. (2021); Cai (2023); Li et al. (2024)). Since our study builds on existing results regarding the approximation capabilities of FNNs, we focus on investigating the approximation abilities of single-layer Transformers in modulating context for ICL tasks. Consequently, our work relies more on the findings from the first category of research. The realization of the UAP depends on the architecture of the network itself, providing constructive insights for exploring the connection between FNNs and Transformers, and offering valuable guidance for our study. Recently, Petrov et al. (2024b) also explored UAP in the context of in-context learning, but without considering vocabulary constraints or positional encodings.

**Transformers.** The Transformer is a widely used neural network architecture for modeling sequences (Vaswani et al. (2017); Devlin et al. (2019); Yang et al. (2019); Raffel et al. (2020); Zhenzhong et al. (2021); Liu et al. (2020)). This non-recurrent architecture relies entirely on the attention mechanism to capture global dependencies between inputs and outputs (Vaswani et al. (2017)). The highly effective neural sequence transduction model is typically structured using an encoder-decoder framework (Bahdanau et al. (2014); Sutskever et al. (2014)). The encoder maps the input sequence $X$ into a continuous representation $S$, from which the decoder generates the output sequence $Y$. In the Transformer, both the encoder and decoder are composed of stacked self-attention layers and fully connected layers. For simplicity, we describe the Transformer using a simplified self-attention sequence encoder. Without positional encoding, the Transformer can be viewed as a stack of $N$ blocks, each consisting of a self-attention layer followed by a feed-forward layer with skip connections. In this paper, we focus on the case of a single-layer self-attention sequence encoder.

**In-context learning.** The Transformer has demonstrated remarkable performance in the field of NLP, and large language models (LLMs) are gaining increasing popularity. ICL has emerged as a new paradigm in NLP, enabling LLMs to make better predictions through prompts provided within the context (Brown et al. (2020); Chowdhery et al. (2023); Touvron et al. (2023); OpenAI et al. (2024); Xun et al. (2017)). We chose ICL as the focus of our research primarily due to its wide range of applications and superior performance, which motivated us to explore its underlying theoretical foundations. ICL delivers high performance with high-quality data at a lower cost (Wang et al. (2021b); Khorashadizadeh et al. (2023); Ding et al. (2023)). It enhances retrieval-augmented methods by prepending grounding documents to the input (Ram et al. (2023)) and can effectively update or refine the model's knowledge base through well-designed prompts (De Cao et al. (2021)).

**Positional Encoding.** The following explanation clarifies the significance of incorporating positional encoding into the Transformer architecture. Recurrent neural networks (RNNs) capture sequential order by encoding the changes in hidden states over time. In contrast, for Transformers, the self-attention mechanism is permutation equivariant, meaning that for any model $f$, any permutation matrix $\pi$, and any input $x$, the following holds: $f(\pi(x)) = \pi(f(x))$. We aim to explore the impact of positional encoding on the performance of a single-layer Transformer when performing ICL tasks with a finite vocabulary. Therefore, we focus on analyzing existing positional encoding methods. There are two fundamental methods for encoding positional information in a sequence within the Transformer: absolute positional encodings (APEs) (*e.g.* He et al. (2021); Liu et al. (2020); Wang et al. (2021a); Ke et al. (2021)), relative positional encodings (RPEs) (*e.g.* Shaw et al. (2018); Dai et al. (2019); Ke et al. (2021)) and rotary positional embedding (RoPE) (Su et al. (2024)). The commonly used APE is implemented by directly adding the positional encodings to the word embeddings, and we follow this implementation.

**UAP of ICL.** Regarding the understanding of the mechanism of ICL, various explanations have been proposed, including those based on Bayesian theory (Xie et al. (2022); Wang et al. (2024)) and gradient descent theory (Dai et al. (2023)). Fine-tuning the Transformer through ICL alters the presentation of the input rather than the model parameters, which is driven by successful few-shot and zero-shot learning (Wei et al. (2022a); Kojima et al. (2022)). This success raises the question of whether we can achieve the UAP through context adjustment.

Yun et al. (2020) demonstrated that Transformers can serve as universal sequence-to-sequence approximators, while Alberti et al. (2023) extended the UAP to architectures with non-standard attention mechanisms. These works represent significant efforts in enabling Transformers to achieve sequence-to-sequence approximation; however, their implementations allow the internal parameters of the Transformers to vary, which does not fully reflect the characteristics of ICL. In contrast, Likhosherstov et al. (2021) showed that while the parameters of self-attention remain fixed, various sparse matrices can be approximated by altering the inputs. Fixing self-attention parameters aligns more closely with practical scenarios and provides valuable insights for our work. However, this approach has the limitation of excluding the full Transformer architecture. Furthermore, Deora et al. (2024) illustrated the convergence and generalization of single-layer multi-head self-attention models trained using gradient descent, supporting the feasibility of our research by emphasizing the robust generalization of Transformers. Nevertheless, Petrov et al. (2024a) indicated that the presence of a prefix does not alter the attention focus within the context, prompting us to explore variations in input context and introduce flexibility in positional encoding.

### 1.3 OUTLINE

We will introduce the notations and background results in Section 2. Section 3 addresses the case where the vocabulary is finite and positional encoding is not used. Section 4 discusses the benefits of using positional encoding. A summary is provided in Section 5. All proof of lemmas and theorems are provided in Appendix.

## 2 BACKGROUND MATERIALS

We consider the approximation problem as follows. For a target continuous function $f : \mathcal{K} \to \mathbb{R}^{d_y}$ with a compact domain $\mathcal{K} \subset \mathbb{R}^{d_x}$, we aim to adjust the content of the context so that the output of the Transformer network can approximate $f$. First, we present the concrete forms and notations for the inputs of ICL, FNNs, and Transformers.

### 2.1 NOTATIONS

**Input of in-context learning.** In the ICL task, the given $n$ demonstrations are denoted as $z^{(i)} = (x^{(i)}, y^{(i)})$ for $i = 1, 2, ..., n$, where $x^{(i)} \in \mathbb{R}^{d_x}$ and $y^{(i)} \in \mathbb{R}^{d_y}$. Unlike the setting in Ahn et al. (2024) and Cheng et al. (2024) where $y^{(i)}$ was related to $x^{(i)}$ (for example $y^{(i)} = \phi(x^{(i)})$ for some function $\phi$), in this paper, we do not assume any correspondence between $x^{(i)}$ and $y^{(i)}$, *i.e.*, $x^{(i)}$ and $y^{(i)}$ are chosen freely. To predict the target at a query vector $x \in \mathbb{R}^{d_x}$ or $z = (x, 0) \in \mathbb{R}^{d_x + d_y}$, we

define the following matrix $Z$ as the input:

$$Z = \begin{bmatrix} z^{(1)} & z^{(2)} & \cdots & z^{(n)} & z \end{bmatrix} := \begin{bmatrix} x^{(1)} & x^{(2)} & \cdots & x^{(n)} & x \\ y^{(1)} & y^{(2)} & \cdots & y^{(n)} & 0 \end{bmatrix} \in \mathbb{R}^{(d_x+d_y)\times(n+1)}. \quad (1)$$

Furthermore, let $\mathcal{P} : \mathbb{N}^+ \to \mathbb{R}^{d_x+d_y}$ represent a positional encoding function, and define $\mathcal{P}^{(i)} := \mathcal{P}(i)$. Denote the demonstrations with positional encoding as $z_{\mathcal{P}}^{(i)} = z^{(i)} + \mathcal{P}^{(i)}$ and $z_{\mathcal{P}} = z + \mathcal{P}^{(n+1)}$. The context with positional encoding can then be represented as:

$$Z_{\mathcal{P}} = \begin{bmatrix} z_{\mathcal{P}}^{(1)} & z_{\mathcal{P}}^{(2)} & \cdots & z_{\mathcal{P}}^{(n)} & z_{\mathcal{P}} \end{bmatrix} := \begin{bmatrix} x_{\mathcal{P}}^{(1)} & x_{\mathcal{P}}^{(2)} & \cdots & x_{\mathcal{P}}^{(n)} & x_{\mathcal{P}} \\ y_{\mathcal{P}}^{(1)} & y_{\mathcal{P}}^{(2)} & \cdots & y_{\mathcal{P}}^{(n)} & y_{\mathcal{P}} \end{bmatrix} \in \mathbb{R}^{(d_x+d_y)\times(n+1)}. \quad (2)$$

Here, the vectors $x_{\mathcal{P}}^{(i)}$ and $y_{\mathcal{P}}^{(i)}$ represent the corresponding components of $z_{\mathcal{P}}^{(i)}$. Additionally, we denote:

$$X = \begin{bmatrix} x^{(1)} & x^{(2)} & \cdots & x^{(n)} \end{bmatrix} \in \mathbb{R}^{d_x \times n}, \quad X_{\mathcal{P}} = \begin{bmatrix} x_{\mathcal{P}}^{(1)} & x_{\mathcal{P}}^{(2)} & \cdots & x_{\mathcal{P}}^{(n)} \end{bmatrix} \in \mathbb{R}^{d_x \times n}, \quad (3)$$

$$Y = \begin{bmatrix} y^{(1)} & y^{(2)} & \cdots & y^{(n)} \end{bmatrix} \in \mathbb{R}^{d_y \times n}, \quad Y_{\mathcal{P}} = \begin{bmatrix} y_{\mathcal{P}}^{(1)} & y_{\mathcal{P}}^{(2)} & \cdots & y_{\mathcal{P}}^{(n)} \end{bmatrix} \in \mathbb{R}^{d_y \times n}. \quad (4)$$

**Feed-forward neural networks.** One-hidden-layer FNNs have sufficient capacity to approximate continuous functions on any compact domain. In this article, all the FNNs we refer to and use are one-hidden-layer networks. We denote a one-hidden-layer FNN with activation function $\sigma$ as $\mathrm{N}^\sigma$, and the set of all such networks is denoted as $\mathcal{N}^\sigma$, *i.e.*,

$$\mathcal{N}^\sigma = \left\{ \mathrm{N}^\sigma := A\,\sigma(Wx+b) \;\middle|\; A \in \mathbb{R}^{d_y \times k}, W \in \mathbb{R}^{k \times d_x}, b \in \mathbb{R}^k, k \in \mathbb{N} \right\} \quad (5)$$

$$= \left\{ \mathrm{N}^\sigma := \sum_{i=1}^{k} a_i \sigma(w_i \cdot x + b_i) \;\middle|\; (a_i, w_i, b_i) \in \mathbb{R}^{d_y} \times \mathbb{R}^{d_x} \times \mathbb{R}, k \in \mathbb{N} \right\}. \quad (6)$$

For elementwise activations, such as ReLU, the above notation is well-defined. However, if the activation function is not elementwise, especially in the case of softmax activation, we need to give more details for the notation:

$$\mathcal{N}^{\mathrm{softmax}} = \left\{ \mathrm{N}^{\mathrm{softmax}} = \frac{\sum_{i=1}^{k} a_i e^{w_i \cdot x + b_i}}{\sum_{i=1}^{k} e^{w_i \cdot x + b_i}} \;\middle|\; (a_i, w_i, b_i) \in \mathbb{R}^{d_y} \times \mathbb{R}^{d_x} \times \mathbb{R}, k \in \mathbb{N} \right\}. \quad (7)$$

**Transformers.** We define the general attention mechanism following Ahn et al. (2024); Cheng et al. (2024) as:

$$\mathrm{Attn}_{Q,K,V}^\sigma(Z) := VZM\sigma((QZ)^\top KZ), \quad (8)$$

where $V, Q, K$ are the value, query, and key matrices in $\mathbb{R}^{(d_x+d_y)\times(d_x+d_y)}$, respectively, $M = \mathrm{diag}(I_n, 0)$ is the mask matrix in $\mathbb{R}^{(n+1)\times(n+1)}$, and $\sigma$ is the activation function. Here the softmax activation of a matrix $G \in \mathbb{R}^{m \times n}$ is defined as:

$$\mathrm{softmax}(G) := \left[ \frac{\exp(G_{i,j})}{\sum_{l=1}^{m} \exp(G_{l,j})} \right]_{i,j}. \quad (9)$$

With this formulation of the general attention mechanism, we can define a single-layer Transformer without positional encoding as:

$$\mathrm{T}^\sigma(x; X, Y) := (Z + VZM\sigma((QZ)^\top KZ))_{d_x+1:d_x+d_y, n+1}, \quad (10)$$

where $[a:b, c:d]$ denotes the submatrix from the $a$-th row to the $b$-th row and from the $c$-th column to the $d$-th column. If $a = b$ (or $c = d$), the row (or column) index is reduced to a single number. Similarly to the notation for FNNs, $\mathcal{T}^\sigma$ denotes the set of all $\mathrm{T}^\sigma$ with different parameters.

**Vocabulary.** In the above notations, the parameters are general and unrestricted. When we refer to a "vocabulary", we mean that the parameters are drawn from a finite set. For networks and their corresponding sets, we use the subscript $*$ to indicate the use of a vocabulary $\mathcal{V}$.

In the context of ICL, we refer to it as vocabulary ICL if all input vectors $z^{(i)}$ come from a finite vocabulary $\mathcal{V} = \mathcal{V}_x \times \mathcal{V}_y \subset \mathbb{R}^{d_x} \times \mathbb{R}^{d_y}$. In this case, we use $\mathrm{T}^\sigma_*(x; X, Y)$ to represent the Transformer $\mathrm{T}^\sigma(x; X, Y)$ defined in equation (10), and denote the set of such Transformers as $\mathcal{T}^\sigma_*$:

$$\mathcal{T}^\sigma_* = \left\{ \mathrm{T}^\sigma_*(x; X, Y) := \mathrm{T}^\sigma(x; X, Y) \,\Big|\, z^{(i)} \in \mathcal{V}, i \in \{1, 2, ..., n\}, n \in \mathbb{N}^+ \right\}. \tag{11}$$

When positional encoding $\mathcal{P}$ is involved, we add the subscript $\mathcal{P}$, i.e.,

$$\mathcal{T}^\sigma_{*,\mathcal{P}} = \left\{ \mathrm{T}^\sigma_{*,\mathcal{P}}(x; X, Y) := \mathrm{T}^\sigma(x; X_\mathcal{P}, Y_\mathcal{P}) \,\Big|\, z^{(i)} \in \mathcal{V}, i \in \{1, 2, ..., n\}, n \in \mathbb{N}^+ \right\}. \tag{12}$$

Note that the context length $n$ in $\mathrm{T}^\sigma$, $\mathrm{T}^\sigma_*$, and $\mathrm{T}^\sigma_{*,\mathcal{P}}$ are unbounded.

For feedforward neural networks (FNNs), we denote a network with a finite set of weights as $\mathrm{N}^\sigma_*$, and the corresponding set of such networks as $\mathcal{N}^\sigma_*$:

$$\mathcal{N}^\sigma_* = \left\{ \mathrm{N}^\sigma_* := \sum_{i=1}^{k} a_i \sigma(w_i \cdot x + b_i) \mid (a_i, w_i, b_i) \in \mathcal{A} \times \mathcal{W} \times \mathcal{B}, k \in \mathbb{N} \right\}. \tag{13}$$

where $\mathcal{A} \subset \mathbb{R}^{d_y}, \mathcal{W} \subset \mathbb{R}^{d_x}$, and $\mathcal{B} \subset \mathbb{R}$ are finite sets.

To simplify calculations and expressions, we introduce the following assumptions throughout the remainder of the article similar to the setting in Cheng et al. (2024).

**Assumption.** *The matrices $Q, K, V \in \mathbb{R}^{(d_x+d_y) \times (d_x+d_y)}$ have the following sparse partition:*

$$Q = \begin{bmatrix} B & 0 \\ 0 & 0 \end{bmatrix}, \quad K = \begin{bmatrix} C & 0 \\ 0 & 0 \end{bmatrix}, \quad V = \begin{bmatrix} D & E \\ F & U \end{bmatrix}, \tag{14}$$

*where $B, C, D \in \mathbb{R}^{d_x \times d_x}$, $E \in \mathbb{R}^{d_x \times d_y}$, $F \in \mathbb{R}^{d_y \times d_x}$ and $U \in \mathbb{R}^{d_y \times d_y}$. We assume the matrices $B, C$ and $U$ are non-singular, and the matrix $F = 0$. In addition, we assume the elementwise activation $\sigma$ is non-polynomial, locally bounded, and continuous.*

We present all our notations in the table below.

Table 1: Table of Notations

| Notations | Explanations |
|---|---|
| $d_x, d_y$ | Dimensions of input and output. |
| $\mathcal{P}$ | Positional encoding. |
| $X, Y$ | Context without positional encoding. |
| $X_\mathcal{P}, Y_\mathcal{P}$ | Context with positional encoding $\mathcal{P}$. |
| $Z$ | Input without positional encoding. |
| $Z_\mathcal{P}$ | Input with positional encoding. |
| $\mathcal{V}$ | Vocabulary of the vectors. |
| $\mathcal{V}_x, \mathcal{V}_y$ | Vocabulary of $x^{(i)}$ and $y^{(i)}$. |
| $\mathrm{N}^\sigma, \mathcal{N}^\sigma$ | One-hidden-layer FNN and its collection. |
| $\mathrm{T}^\sigma, \mathcal{T}^\sigma$ | Single-layer Transformer and its collection. |
| $\mathrm{N}^\sigma_*, \mathcal{N}^\sigma_*$ | One-hidden-layer FNN with a finite set of weights and its collection. |
| $\mathrm{T}^\sigma_*, \mathcal{T}^\sigma_*$ | Single-layer Transformer with vocabulary restrictions and its collection. |
| $\mathrm{T}^\sigma_{*,\mathcal{P}}, \mathcal{T}^\sigma_{*,\mathcal{P}}$ | Single-layer Transformer with positional encoding, vocabulary restrictions, and its collection. |
| $\| \cdot \|$ | The uniform norm of vectors, *i.e.*, a shorthand for $\| \cdot \|_\infty$. |
| $\tilde{x}$ | Append a one to the end of $x$, *i.e.*, $\tilde{x} = \begin{bmatrix} x \\ 1 \end{bmatrix}$. |

## 2.2 UNIVERSAL APPROXIMATION PROPERTY

The vanilla form of the universal approximation property for feedforward neural networks plays a crucial role in our study. We state it in the following lemma:

**Lemma 1** (UAP of FNNs (Leshno et al. (1993))). *Let $\sigma : \mathbb{R} \to \mathbb{R}$ be a non-polynomial, locally bounded, piecewise continuous activation function. For any continuous function $f : \mathbb{R}^{d_x} \to \mathbb{R}^{d_y}$ defined on a compact domain $\mathcal{K}$, and for any $\varepsilon > 0$, there exist $k \in \mathbb{N}$, $A \in \mathbb{R}^{d_y \times k}$, $b \in \mathbb{R}^k$, and $W \in \mathbb{R}^{k \times d_x}$ such that*

$$\|A\sigma(Wx + b) - f(x)\| < \varepsilon, \quad \forall x \in \mathcal{K}. \tag{15}$$

The theorem presented above is well-known and primarily applies to activation functions operating pointwise. However, it can be readily extended to the case of the softmax activation function. In fact, this can be achieved using neural networks with exponential activation functions. The specific approach for this generalization is detailed in Appendix A.

## 2.3 FEED-FORWARD NEURAL NETWORKS AND TRANSFORMERS

It is important to emphasize the connection between FNNs and Transformers.

**Lemma 2.** *Let $\sigma$ be an elementwise activation and $\mathrm{T}^\sigma$ be a single-layer Transformer. For any one-hidden-layer network $\mathrm{N}^\sigma : \mathbb{R}^{d_x-1} \to \mathbb{R}^{d_y} \in \mathcal{N}^{\mathrm{ReLU}}$ with $n$ hidden neurons, there exist matrices $X \in \mathbb{R}^{d_x \times n}$ and $Y \in \mathbb{R}^{d_y \times n}$ such that*

$$\mathrm{T}^\sigma(\tilde{x}; X, Y) = \mathrm{N}^\sigma(x), \quad \forall x \in \mathbb{R}^{d_x-1}. \tag{16}$$

There is a difference in the input dimensions of $\mathrm{T}^\sigma$ and $\mathrm{N}^\sigma$, as the latter includes a bias dimension absent in the former. To connect the two inputs, $\tilde{x}$ and $x$, we use a tilde, where $\tilde{x}$ is formed by augmenting $x$ with an additional one appended to the end.

By employing the structure of query, key, and value matrices in (14), the output forms of the Transformer $\mathrm{T}^\sigma(\tilde{x}; X, Y)$ can be simplified as follows:

$$\mathrm{T}^\sigma(\tilde{x}; X, Y) = \left( \begin{bmatrix} X & \tilde{x} \\ Y & 0 \end{bmatrix} + \begin{bmatrix} DX + EY & 0 \\ FX + UY & 0 \end{bmatrix} \sigma \left( \begin{bmatrix} X^\top B^\top CX & X^\top B^\top C\tilde{x} \\ \tilde{x}^\top B^\top CX & \tilde{x}^\top B^\top C\tilde{x} \end{bmatrix} \right) \right)_{d_x+1:d_x+d_y, n+1}$$

$$= (FX + UY)\sigma(X^\top B^\top C\tilde{x}) = UY\sigma(X^\top B^\top C\tilde{x}). \tag{17}$$

Comparing this with the output form of FNNs, $\mathrm{N}^\sigma(x) = A\sigma(Wx + b)$, it becomes evident that setting $X = (C^\top B)^{-1} \begin{bmatrix} W & b \end{bmatrix}^\top$ and $Y = U^{-1}A$ is sufficient to finish the proof.

It can be observed that the form in equation (17) exhibits the structure of an FNN. Consequently, Lemma 2 implies that single-layer Transformers $\mathrm{T}^\sigma$ with in-context learning and FNNs $\mathrm{N}^\sigma$ are equivalent. However, this equivalence does not hold for the case of softmax activation due to differences in the normalization operations between FNNs and Transformers. Therefore, in the subsequent sections of this article, we employ different analytical methods to address the two types of activation functions.

Moreover, the equivalence in equation (31) suggests that the context in Transformers can act as a control parameter for the model, thereby endowing it with the universal approximation property. This offers a novel perspective on the parameterization of FNNs.

## 2.4 UNIVERSAL APPROXIMATION PROPERTY OF IN-CONTEXT LEARNING

We now present the UAP of Transformers in the context of ICL.

**Lemma 3.** *Let $\mathrm{T}^\sigma$ be a single-layer Transformer with elementwise or softmax activation, and $\mathcal{K}$ be a compact domain in $\mathbb{R}^{d_x-1}$. Then for any continuous function $f : \mathcal{K} \to \mathbb{R}^{d_y}$ and any $\varepsilon > 0$, there exist matrices $X \in \mathbb{R}^{d_x \times n}$ and $Y \in \mathbb{R}^{d_y \times n}$ such that*

$$\|\mathrm{T}^\sigma(\tilde{x}; X, Y) - f(x)\| < \varepsilon, \quad \forall x \in \mathcal{K}. \tag{18}$$

For the case of elementwise activation, the result follows directly by combining Lemma 1 and Lemma 2. However, for the softmax activation, the normalization operation requires an additional

technique in the proof. The key idea is to consider an FNN with the exponential function as its activation and introduce an additional neuron to account for the normalization effect. Detailed proofs are provided in Appendix A. Similar results have also been reported in recent work Petrov et al. (2024b), albeit using different techniques.

# 3 THE NON-UNIVERSAL APPROXIMATION PROPERTY OF $\mathcal{N}_*^\sigma$ AND $\mathcal{T}_*^\sigma$

One key aspect of ICL is that the context can act as a control parameter for the model. We now consider the case where the context is restricted to a finite vocabulary. A natural question arises: can a single-layer Transformer with a finite vocabulary, $\mathrm{T}_*^\sigma \in \mathcal{T}_*^\sigma$, still achieve the UAP? Given the established connection between FNNs and Transformers, we first analyze $\mathrm{N}_*^\sigma \in \mathcal{N}_*^\sigma$ for simplicity.

The answer is that $\mathcal{N}_*^\sigma$ cannot achieve the UAP because the parameters can only take on a finite number of values. For elementwise activations, the span of $\mathcal{N}_*^\sigma$, span($\mathcal{N}_*^\sigma$), forms a finite-dimensional function space. According to results from functional analysis, $\mathcal{N}_*^\sigma$ is closed under the function norm (see e.g. Theorem 1.21 of Rudin (1991) or Corollary C.4 of Cannarsa & D'Aprile (2015)). This implies that the set of functions approximable by span($\mathcal{N}_*^\sigma$) is precisely the set of functions within span($\mathcal{N}_*^\sigma$). Consequently, any function not in span($\mathcal{N}_*^\sigma$) cannot be arbitrarily approximated, meaning that the UAP cannot be achieved.

For softmax networks, the normalization operation introduces further limitations. Even though $\mathrm{N}_*^{\mathrm{softmax}}$ consists of weighted units drawn from a fixed finite collection of basic units, normalization prevents these networks from being simple linear combinations of one another. While the span of $\mathcal{N}_*^{\mathrm{softmax}}$ might theoretically have infinite dimensionality, its expressive power remains constrained.

To better understand the behavior of functions within $\mathcal{N}_*^{\mathrm{softmax}}$, we present the following proposition as an introduction.

**Proposition 4.** *The scalar function* $h_k(x) = \sum\limits_{i=1}^{k} a_i e^{b_i x}$, *where* $a_i, b_i, x \in \mathbb{R}$ *and at least one* $a_i$ *is nonzero, has at most* $k-1$ *zero points.*

The function $h_k(x)$ is commonly referred to as a sum of exponentials. Proposition 4 establishes the maximum number of zero points for this class of functions. The result can be proved using mathematical induction. The cases for $k = 1$ and $k = 2$ are straightforward. Assuming the proposition holds for $k = N$, we proceed with a proof by contradiction for $k = N + 1$. Assume $a_{N+1} \neq 0$ and $h(x)$ has $N + 1$ zero points. We can define a new function $g$ that shares the same zero points as $h_{N+1}$, given by

$$g(x) = \frac{h_k(x)}{a_{N+1} e^{b_{N+1} x}} = 1 + \sum_{i=1}^{N} \frac{a_i}{a_{N+1}} e^{(b_i - b_{N+1})x}. \tag{19}$$

The derivative of $g$ is the sum of $N$ exponentials. By applying the intermediate value theorem, we show that if the number of zero points exceeds $N$, it leads to a contradiction.

As a consequence of Proposition 4, we know that a sum of $k$ exponential functions cannot arbitrarily approximate certain functions, such as $f(x) = \sin((k+1)\pi x)$ over the interval $[0, 2]$. The function $f(x)$ has $k + 1$ peaks and $k + 1$ zeros within this interval. By applying the intermediate value theorem, we conclude that any function approximating $f(x)$ closely must also exhibit more than $k$ zeros, leading to a contradiction. This limitation in the approximation power of sums of exponentials extends naturally to multivariate functions and applies to softmax activations, where the normalization further restricts expressiveness.

Now we can summarize the non-universal approximation property of $\mathcal{N}_*^\sigma$ in the following lemma.

**Lemma 5.** *The function class* $\mathcal{N}_*^\sigma$, *with elementwise or softmax activation* $\sigma$, *cannot achieve the UAP. Specifically, for any compact domain* $\mathcal{K} \subset \mathbb{R}^{d_x}$, *there exists a continuous function* $f : \mathcal{K} \to \mathbb{R}^{d_y}$ *and* $\varepsilon_0 > 0$ *such that* $\max\limits_{x \in \mathcal{K}} \|f(x) - \mathrm{N}_*^\sigma(\tilde{x})\| \geq \varepsilon_0$ *for all* $\mathrm{N}_*^\sigma \in \mathcal{N}_*^\sigma$.

By leveraging the connection between FNNs and Transformers, we establish Theorem 6.

**Theorem 6.** *The function class* $\mathcal{T}_*^\sigma$, *with elementwise or softmax activation* $\sigma$, *cannot achieve the UAP. Specifically, for any compact domain* $\mathcal{K} \subset \mathbb{R}^{d_x - 1}$, *there exists a continuous function* $f : \mathcal{K} \to$

$\mathbb{R}^{d_y}$ *and* $\varepsilon_0 > 0$ *such that*

$$\max_{x \in \mathcal{K}} \|f(x) - \mathrm{T}_*^\sigma(\tilde{x})\| \geq \varepsilon_0, \quad \forall\, \mathrm{T}_*^\sigma \in \mathcal{T}_*^\sigma. \tag{20}$$

The result for elementwise activations follows directly from the application of Lemma 2 and Lemma 5. However, the case of the softmax activation is more intricate, as it requires additional techniques to account for the normalization effect. The proof, which utilizes Proposition 4 once again, is presented in the Appendix B.

It is worth noting that Theorem 6 holds even without imposing any constraints on the value, query, and key matrices, $V$, $Q$, and $K$ (e.g., the sparse partition described in equation (14)). For further details, refer to Appendix D.

## 4 THE UNIVERSAL APPROXIMATION PROPERTY OF $\mathcal{T}_{*,\mathcal{P}}^\sigma$

After establishing that neither $\mathcal{N}_*^\sigma$ nor $\mathcal{T}_*^\sigma$ can achieve the UAP, we aim to leverage a key feature of Transformers: their ability to incorporate absolute positional encodings during token input. This motivates us to investigate whether $\mathcal{T}_{*,\mathcal{P}}^\sigma$ can realize the UAP.

To facilitate our constructive proof, we introduce Lemma 7 as an auxiliary tool to support the main theorem.

**Lemma 7** (Kronecker Approximation Theorem (see e.g. Apostol (1989))). *Given real $n$-tuples* $\alpha^{(i)} = (\alpha_1^{(i)}, \alpha_2^{(i)}, \cdots, \alpha_n^{(i)}) \in \mathbb{R}^n$ *for* $i = 1, \cdots, m$ *and* $\beta = (\beta_1, \beta_2, \cdots, \beta_n) \in \mathbb{R}^n$, *the following condition holds: for any* $\varepsilon > 0$, *there exist* $q_i, l_i \in \mathbb{Z}$ *such that*

$$\left\| \beta_j - \sum_{i=1}^m q_i \alpha_j^{(i)} + l_j \right\| < \varepsilon, \quad 1 \leq j \leq n, \tag{21}$$

*if and only if for any* $r_1, \cdots, r_n \in \mathbb{Z}, i = 1, \cdots, m$ *with*

$$\sum_{j=1}^n \alpha_j^{(i)} r_j \in \mathbb{Z}, \quad i = 1, \cdots, m, \tag{22}$$

*the number* $\sum_{j=1}^n \beta_j r_j$ *is also an integer. In the case of* $m = 1$ *and* $n = 1$, *for any* $\alpha, \beta, \varepsilon \in \mathbb{R}$ *with* $\alpha$ *irrational and* $\varepsilon > 0$, *there exist integers* $l$ *and* $q$ *with* $q > 0$ *such that* $|\beta - q\alpha + l| < \varepsilon$.

This lemma (Lemma 7) indicates that if the condition in equation (22) is satisfied only when all $r_i$ are zeros, then the set $\{Mq + l \mid q \in \mathbb{Z}^m, l \in \mathbb{R}^n\}$ is dense in $\mathbb{R}^n$, where the matrix $M \in \mathbb{R}^{n \times m}$ is assembled with vectors $\alpha^{(i)}$, *i.e.*, $M = [\alpha^{(1)}, \alpha^{(2)}, ..., \alpha^{(m)}]$ In the case of $m = n = 1$, let $\alpha = \sqrt{2}$. Then, Lemma 7 implies that the set $\{q\sqrt{2} \pm l \mid l \in \mathbb{N}^+, q \in \mathbb{N}^+\}$ is dense in $\mathbb{R}$. We will build upon this result to prove one of the most significant theorems in this article.

**Theorem 8.** *Let* $\mathcal{T}_{*,\mathcal{P}}^\sigma$ *be the class of functions* $\mathrm{T}_{*,\mathcal{P}}^\sigma$, *where* $\sigma$ *is an elementwise activation, the subscript refers the finite vocabulary* $\mathcal{V} = \mathcal{V}_x \times \mathcal{V}_y$, $\mathcal{P} = \mathcal{P}_x \times \mathcal{P}_y$ *represents the positional encoding map, and denote the set* $S$ *as:*

$$S := \mathcal{V}_x + \mathcal{P}_x = \left\{ x_i + \mathcal{P}_x^{(j)} \ \middle|\ x_i \in \mathcal{V}_x, i, j \in \mathbb{N}^+ \right\}. \tag{23}$$

*If* $S$ *is dense in* $\mathbb{R}^{d_x}$, $\{1, -1, \sqrt{2}, 0\}^{d_y} \subset \mathcal{V}_y$ *and* $\mathcal{P}_y = 0$, *then* $\mathcal{T}_{*,\mathcal{P}}^\sigma$ *can achieve the UAP. That is, for any continuous function* $f : \mathbb{R}^{d_x - 1} \to \mathbb{R}^{d_y}$ *defined on a compact domain* $\mathcal{K}$, *and for any* $\varepsilon > 0$, *there always exist* $X \in \mathbb{R}^{d_x \times n}$ *and* $Y \in \mathbb{R}^{d_y \times n}$ *from the vocabulary* $\mathcal{V}$ *(i.e.,* $x^{(i)} \in \mathcal{V}_x, y^{(i)} \in \mathcal{V}_y$*) with some length* $n \in \mathbb{N}^+$ *such that*

$$\left\| \mathrm{T}_{*,\mathcal{P}}^\sigma\left(\tilde{x}; X, Y\right) - f(x) \right\| < \varepsilon, \quad \forall x \in \mathcal{K}. \tag{24}$$

We provide a constructive proof in Appendix C, and here we only demonstrate the proof idea by considering the specific case of $d_y = 1$ and assuming the matrices $U$, $B$, $C$, and $D$ in the Transformer are identity matrices. In this case, the Transformer $\mathrm{T}_{*,\mathcal{P}}^\sigma(\tilde{x}; X, Y)$ can be simplified to an

FNN, $N_*^\sigma$, similar to the calculation in equation (17):

$$N_*^\sigma(x) = Y\sigma\left(X_\mathcal{P}^\top \tilde{x}\right) = \sum_{j=1}^{n} y^{(j)}\sigma\left(\left(x^{(j)} + \mathcal{P}_x^{(j)}\right)\cdot\tilde{x}\right). \tag{25}$$

The UAP of FNNs shown in Lemma 1 implies that the target function $f$ can be approximated by an FNN $N^\sigma(x)$ with $k$ hidden neurons:

$$N^\sigma(x) = A\sigma(W\tilde{x}) = \sum_{i=1}^{k} a_i\sigma(w_i\cdot\tilde{x}). \tag{26}$$

Since we are considering a continuous activation function $\sigma$, we can conclude that slightly perturbing the parameters $A$ and $W$ will lead to new FNNs that can still approximate $f$, provided the perturbations are small enough. This observation motivates us to construct a proof using the property that each $w_i \in \mathbb{R}^{d_x}$ can be approximated by vectors in $S$, and each $a_i \in \mathbb{R}$ can be approximated by numbers of the form $q_i\sqrt{2} \pm l_i$, with positive integers $q_i$ and $l_i$. Note that the summation $\sum_{i=1}^{k}(q_i\sqrt{2} \pm l_i)\sigma(w_i\cdot\tilde{x})$ can be reformulated as $\sum_{i'=1}^{k'} y_{i'}\sigma(w_{i'}\cdot\tilde{x})$ with $k' = \sum_{i}^{k}(q_i + l_i), y_{i'} \in \{\sqrt{2}, \pm 1\}$ and $w_{i'} \in \{w_1, ..., w_k\}$. For each $w_{i'}$, we can choose a vector $\hat{w}_{i'} := x_{j_{i'}} + \mathcal{P}_x^{(j_{i'})} \in S$ that approximates $w_{i'}$ well, where $j_{i'} \in \mathbb{N}^+$ and $x_{j_{i'}} \in \mathcal{V}_x$. The integers $j_{i'}$ can be chosen to be distinct from each other.

Now, the FNN in (25) can be constructed by using $n = \max(j_1, j_2, \ldots, j_{k'})$ neurons, where the $j$-th neuron is assigned by setting $y^{(j)} = y_{i'} \in \mathcal{V}_y$ and $x^{(j)} = x_{j_{i'}} \in \mathcal{V}_x$ for the case of $j = j_{i'} \in \{j_1, j_2, \ldots, j_{k'}\}$, and $y^{(j)} = 0 \in \mathcal{V}_y$ for the case of $j \notin \{j_1, j_2, \ldots, j_{k'}\}$. Here, the nonzero value of $y^{(j)}$ highlights useful positions and demonstrations.

In the proof idea above, we take the density of the set $S$ in $\mathbb{R}^{d_x}$ as a fundamental assumption. $\mathcal{V}_x$ contains only finitely many elements, rendering it bounded. For $S = \mathcal{V}_x + \mathcal{P}_x$ to be dense in the entire space, $\mathcal{P}_x$ must be unbounded. Next, we relax this requirement, eliminating the need for $\mathcal{P}_x$ to be unbounded, making the conditions more aligned with practical scenarios. Particularly, we consider the specific activation function in the following Theorem 9, where the notations not explicitly mentioned remain consistent with those in Theorem 8.

**Theorem 9.** *If the set $S$ is dense in $[-1, 1]^{d_x}$, then $\mathcal{T}_{*,\mathcal{P}}^{\mathrm{ReLU}}$ is capable of achieving the UAP. Additionally, if $S$ is only dense in a neighborhood $B(w^*, \delta)$ of a point $w^* \in \mathbb{R}^{d_x}$ with radius $\delta > 0$, then the class of transformers with exponential activation, $\mathcal{T}_{*,\mathcal{P}}^{\exp}$, is capable of achieving the UAP.*

The density condition on $S$ is significantly refined here. This improvement is possible because the proof of Theorem 8 relies directly on the UAP of FNNs, where the weights take values from the entire parameter space. However, for FNNs with specific activations, we can restrict the weights to a small set without losing the UAP.

For ReLU networks, we can use the positive homogeneity property, *i.e.*, $A\mathrm{ReLU}(W\tilde{x}) = \frac{1}{\lambda}A\mathrm{ReLU}(\lambda W\tilde{x})$ for any $\lambda > 0$, to restrict the weight matrix $W$. In fact, the restriction that all elements of $W$ take values in the interval $[-1, 1]$ does not affect the UAP of ReLU FNNs because the scale of $W$ can be recovered by adjusting the scale of $A$ via choosing a proper $\lambda$.

For exponential networks, the condition on $S$ is much weaker than in the ReLU case. This relaxation is nontrivial, and the proof stems from a property of the derivatives of exponential functions. Consider the exponential function $\exp(w\cdot x)$ as a function of $w \in B(w^*, \delta)$, and denote it as $h(w)$,

$$h(w) = \exp(w\cdot x) \equiv \exp(w_1 x_1 + \cdots + w_d x_d), \quad w, x \in \mathbb{R}^d, \quad d = d_x, \tag{27}$$

where $w_i$ and $x_i \in \mathbb{R}$ are the components of $w$ and $x$, respectively. Calculating the partial derivatives of $h(w)$, we observe the following relations:

$$\frac{\partial^\alpha h}{\partial w^\alpha} \equiv \frac{\partial^{|\alpha|} h}{\partial w_1^{\alpha_1}\cdots\partial w_d^{\alpha_d}} = x_1^{\alpha_1}\cdots x_d^{\alpha_d} h(w), \tag{28}$$

where $\alpha = (\alpha_1, \ldots, \alpha_d) \in \mathbb{N}^d$ is the index vector representing the order of partial derivatives, and $|\alpha| := \alpha_1 + \cdots + \alpha_d$. This relationship allows us to link exponential FNNs to polynomials since

any polynomial $P(x)$ can be represented in the following form:

$$P(x) = \exp(-w^* \cdot x) \left( \sum_{\alpha \in \Lambda} a_\alpha \frac{\partial^{|\alpha|} h}{\partial w^\alpha} \right)\Bigg|_{w=w^*}, \tag{29}$$

where $a_\alpha$ are the coefficients of the polynomials, $\Lambda$ is a finite set of indices, and the partial derivatives can be approximated by finite differences, which are FNNs. For example, the first-order partial derivative $\frac{\partial h}{\partial w_1}\big|_{w=w^*} = x_1 h(w^*)$ can be approximated by the following difference with a small nonzero number $\lambda \in (0, \delta)$,

$$\frac{h(w^* + \lambda e_1) - h(w^*)}{\lambda} = \frac{1}{\lambda} \exp((w^* + \lambda e_1) \cdot x) - \frac{1}{\lambda} \exp(w^* \cdot x). \tag{30}$$

This is an exponential FNN with two neurons. Finally, employing the well-known Stone-Weierstrass theorem, which states that any continuous function $f$ on compact domains can be approximated by polynomials, and combining the above relations between FNNs and polynomials, we can establish the UAP of exponential FNNs with weight constraints.

**Remark 10.** *When discussing density, one of the most immediate examples that comes to mind is the density of rational numbers in $\mathbb{R}$. How can we effectively enumerate rational numbers? The work by Calkin & Wilf (2000) introduces an elegant method for enumerating positive rational numbers, synthesizing ideas from Stern (1858) and Berndt et al. (1990). It demonstrates the computational feasibility of enumeration through an effective algorithm. Thus, we assume that positional encodings can be implemented using computer algorithms, such as iterative functions.*

## 5 CONCLUSION

In this paper, we establish a connection between feedforward neural networks and Transformers through in-context learning. By leveraging the universal approximation property of FNNs, we demonstrate that the UAP of in-context learning holds when the context is selected from the entire vector space. When the context is drawn from a finite set, we explore the approximation power of vocabulary-based in-context learning, showing that the UAP is achievable only when appropriate positional encodings are incorporated, underscoring the importance of positional encodings.

In our work, we consider Transformers with input sequences of arbitrary length, implying that the positional encoding $\mathcal{P}_x$ consists of a countably infinite set of elements, independent of the target function. As a result, the set $S$ is also infinitely large and may or may not be dense in $\mathbb{R}^d$. In Theorem 8, we assume a strong density condition, which is later relaxed in Theorem 9. However, in practical applications, input sequences are finite, typically truncated for computational feasibility. This shift allows our conclusions to be interpreted through an approximation lens, where the objective is to approximate functions within a specified error margin, rather than achieving infinitesimal precision. Additionally, to achieve universal approximation, it is insightful to compare the function approximation capabilities of our approach (outlined in Lemma 3) with the direct use of FNNs, particularly when the Transformer parameters are trainable.

It is important to note that this paper is limited to single-layer Transformers with absolute positional encodings, and the main results (Theorem 8 and Theorem 9) focus on elementwise activations. Future research should extend these findings to multi-layer Transformers, general positional encodings (such as RPEs and RoPE), and softmax activations. For softmax Transformers, our analysis in Sections 2 and 3 highlighted their connection to Transformers with exponential activations. However, extending this connection to the scenario in Section 4 proves challenging and requires more sophisticated techniques.

Although this paper primarily addresses theoretical issues, we believe our results can offer valuable insights for practitioners. Specifically, in Remark 10, we observe that certain algorithms use function composition to enumerate numbers dense in $\mathbb{R}$. This idea could inspire the design of positional encodings via compositions of fixed functions, similar to RNN approaches. RNNs capture the sequential nature of information by integrating the importance of word order in sentence meaning. However, to the best of our knowledge, existing research on RNNs has not explored the denseness properties of the sets formed by their hidden state sequences. We hope this unexplored property will inspire experimental research in future studies. Furthermore, our construction for Theorem 8 relies on the sparse partition assumption in equation (14). The practical validity of this assumption remains uncertain, and we leave this question open for future exploration.

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

## A    PROOF FOR SECTION 2

We will lay out some lemmas mentioned in this article below. In this part of our Appendix, we consider a more general case, of which $E$ and $F$ not zero matrixes.

### A.1    PROOF OF LEMMA 2

**Lemma 2.** *Let $\sigma$ be an elementwise activation and $\mathrm{T}^\sigma$ be a single-layer Transformer. For any one-hidden-layer network $\mathrm{N}^\sigma : \mathbb{R}^{d_x-1} \to \mathbb{R}^{d_y} \in \mathcal{N}^{\mathrm{ReLU}}$ with $n$ hidden neurons, there exist matrices $X \in \mathbb{R}^{d_x \times n}$ and $Y \in \mathbb{R}^{d_y \times n}$ such that*

$$\mathrm{T}^\sigma\left(\tilde{x}; X, Y\right) = \mathrm{N}^\sigma(x), \quad \forall x \in \mathbb{R}^{d_x-1}. \tag{31}$$

*Proof.* We can directly compute the following

$$
\begin{aligned}
&\mathrm{T}^{\mathrm{ReLU}}\left(\tilde{x}; X, Y\right) \\
&= \left(Z + \mathrm{Attn}_{Q,K,V}^{\mathrm{ReLU}}(\tilde{x}; X, Y)\right)_{d_x+1:d_x+d_y, n+1} \\
&= \left(Z + VZM\,\mathrm{ReLU}(Z^\top Q^\top K Z)\right)_{d_x+1:d_x+d_y, n+1} \\
&= \left(Z + \begin{bmatrix} DX + EY & 0 \\ FX + UY & 0 \end{bmatrix} \begin{bmatrix} \mathrm{ReLU}(X^\top B^\top C X) & \mathrm{ReLU}(X^\top B^\top C \tilde{x}) \\ \mathrm{ReLU}(\tilde{x}^\top B^\top C X) & \mathrm{ReLU}(\tilde{x}^\top B^\top C \tilde{x}) \end{bmatrix}\right)_{d_x+1:d_x+d_y, n+1}.
\end{aligned} \tag{32}
$$

It is obvious that

$$\mathrm{T}^{\mathrm{ReLU}}(\tilde{x}; X, Y) = (FX + UY)\,\mathrm{ReLU}(X^\top B^\top C \tilde{x}). \tag{33}$$

Assume $\mathrm{N}^{\mathrm{ReLU}}(x) = A\,\mathrm{ReLU}(Wx + b)$ is an arbitrary single-layer FNN, where $W \in \mathbb{R}^{k \times d_x}, A \in \mathbb{R}^{d_y \times k}, b \in \mathbb{R}^k$, and $k$ represents the width of hidden layer.

Let us set the length of context to $k$, that is $X \in \mathbb{R}^{d_x \times k}, Y \in \mathbb{R}^{d_y \times k}$. Through trivial calculation we can find that if we set

$$X = (CB)^{-1} \begin{bmatrix} W^\top \\ b^\top \end{bmatrix}, \quad Y = U^{-1}(A - FX), \tag{34}$$

then $\mathcal{T}^{\mathrm{ReLU}}\left(\tilde{x}; X, Y\right) = \mathcal{N}^{\mathrm{ReLU}}(x)$ holds. $\qquad\square$

### A.2    PROOF OF THE UAP OF SOFTMAX FNNs

**Lemma 11.** *For any continuous function $f : \mathbb{R}^{d_x} \to \mathbb{R}^{d_y}$ defined on a compact domain $\mathcal{K}$ and $\varepsilon > 0$, there always exist a softmax FNN $\mathrm{N}^{\mathrm{softmax}}(x) : \mathbb{R}^{dx} \to \mathbb{R}^{d_y}$ satisfying*

$$\|\mathrm{N}^{\mathrm{softmax}}(x) - f(x)\| < \varepsilon. \tag{35}$$

*Proof.* According to Lemma 1 we can construct a network

$$
\begin{aligned}
\mathrm{N}^{\exp}(x) &= A \exp(Wx + b) \\
&= A \begin{bmatrix} \exp((Wx+b)_1) \\ \exp((Wx+b)_2) \\ \cdots \\ \exp((Wx+b)_k) \end{bmatrix}
\end{aligned} \tag{36}
$$

such that $\|\mathrm{N}^{\exp}(x) - f(x)\| < \varepsilon$ for all $x \in \mathcal{K}$ and $k$ represents the width of hidden layer. We now construct a softmax network as follows

$$\mathrm{N}^{\mathrm{softmax}}(x) = A'\,\mathrm{softmax}\left(\begin{bmatrix} Wx + b' \\ 0 \end{bmatrix}\right), \tag{37}$$

where every element in $b' = b'(\varepsilon)$ is sufficiently small to satisfy $\exp((W_1 x + b')_i) < \frac{\varepsilon'}{k}$ for all $x \in \mathcal{K}, i = 1, 2, \cdots, k$, and $A'_{i,j} = \begin{cases} A_{i,j} \exp(b_j - b'_j) & j = 1, \cdots, k \\ 0 & j = k+1 \end{cases}$, where $i = 1, \cdots, d_y$. We can compute that

$$\|f(x) - \mathrm{N}^{\mathrm{softmax}}(x)\| \leq \|f(x) - \mathrm{N}^{\exp}(x)\| + \|\mathrm{N}^{\exp}(x) - \mathrm{N}^{\mathrm{softmax}}(x)\|. \tag{38}$$

We focus on estimating of the upper bound of the second term, since it is evident that the first term does not exceed $\varepsilon$.

$$
\begin{aligned}
\|\mathrm{N}^{\exp} - \mathrm{N}^{\mathrm{softmax}}(x)\| &\leq \max_{1 \leq i \leq d_y} \left\{ \left| \sum_{j=1}^{k} A_{i,j} \exp((Wx+b)_j) - \frac{\sum_{j=1}^{k} A'_{i,j} \exp((Wx+b')_j)}{1 + \sum_{j=1}^{k} \exp((Wx+b')_j)} \right| \right\} \\
&= \max_{1 \leq i \leq d_y} \left\{ \left| \sum_{j=1}^{k} A_{i,j} \exp((Wx+b)_j) - \frac{\sum_{j=1}^{k} A_{i,j} \exp((Wx+b)_j)}{1 + \sum_{j=1}^{k} \exp((Wx+b')_j)} \right| \right\} \\
&\leq \|\mathrm{N}^{\exp}(x)\| \left( 1 - \frac{1}{1 + \sum_{j=1}^{k} \exp((Wx+b')_j)} \right) \\
&\leq \|\mathrm{N}^{\exp}(x)\| \left( 1 - \frac{1}{1 + \varepsilon'} \right). \\
&\leq \|\mathrm{N}^{\exp}(x)\| \varepsilon'.
\end{aligned}
\tag{39}
$$

By setting $\varepsilon' = \frac{\varepsilon}{\|\mathrm{N}^{\exp}(x)\|}$, we ensure it is finite, leading to the conclusion that

$$
\|f(x) - \mathrm{N}^{\mathrm{softmax}}(x)\| \leq 2\varepsilon.
\tag{40}
$$

$\square$

### A.3 PROOF OF LEMMA 3

**Lemma 3.** *Let $\mathrm{T}^{\sigma}$ be a single-layer Transformer with elementwise or softmax activation, and $\mathcal{K}$ be a compact domain in $\mathbb{R}^{d_x-1}$. Then for any continuous function $f : \mathcal{K} \to \mathbb{R}^{d_y}$ and any $\varepsilon > 0$, there exist matrices $X \in \mathbb{R}^{d_x \times n}$ and $Y \in \mathbb{R}^{d_y \times n}$ such that*

$$
\|\mathrm{T}^{\sigma}(\tilde{x}; X, Y) - f(x)\| < \varepsilon, \quad \forall x \in \mathcal{K}.
\tag{41}
$$

*Proof.* For ReLU case, with the help of Lemma 1 and 2, the conclusion follows trivially.

Then we solve the softmax case. Similarly, for any $\varepsilon > 0$, we can construct an exponential FNN $\mathrm{N}^{\mathrm{softmax}}(x) = A \operatorname{softmax}\left( \begin{bmatrix} Wx+b \\ 0 \end{bmatrix} \right)$ using Lemma 3 such that $\|\mathrm{N}^{\mathrm{softmax}} - f(x)\| < \varepsilon$ and it has $k$ hidden neurons. What we need to do is to approximate this softmax FNN with a softmax Transformer. We can directly compute the following

$$
\begin{aligned}
&\mathrm{T}^{\mathrm{softmax}}(\tilde{x}; X, Y) \\
&= \left( Z + \mathrm{Attn}_{Q,K,V}^{\mathrm{softmax}}(\tilde{x}; X, Y) \right)_{d_x+1:d_x+d_y, n+1} \\
&= \left( Z + VZM \operatorname{softmax}(Z^{\top} Q^{\top} K Z) \right)_{d_x+1:d_x+d_y, n+1} \\
&= \left( Z + \begin{bmatrix} DX+EY & 0 \\ FX+UY & 0 \end{bmatrix} \operatorname{softmax}\left( \begin{bmatrix} X^{\top}B^{\top}CX & X^{\top}B^{\top}C\tilde{x} \\ \tilde{x}^{\top}B^{\top}CX & \tilde{x}^{\top}B^{\top}C\tilde{x} \end{bmatrix} \right) \right)_{d_x+1:d_x+d_y, n+1}.
\end{aligned}
\tag{42}
$$

It is obvious that

$$
\mathrm{T}^{\mathrm{softmax}}(x; X, Y) = (FX+UY) \operatorname{softmax}\left( \begin{bmatrix} X^{\top}B^{\top}C\tilde{x} \\ \tilde{x}^{\top}B^{\top}C\tilde{x} \end{bmatrix} \right)_{1:n}.
\tag{43}
$$

Then through comparing the output of the softmax Transformer with the exponential FNN, we can find out that there is one more bounded positive term $t(x) = \exp(\tilde{x}^{\top}B^{\top}C\tilde{x})$ when processing

normalization. Assume $X^\top B^\top C = \begin{bmatrix} W & b+s\mathbf{1} \\ 0 & s \end{bmatrix} \in \mathbb{R}^{(k+1)\times(d_x)}$, $FX + UY = [A \quad 0] \in \mathbb{R}^{d_y\times(k+1)}$, where $s = s(\varepsilon')$ is big enough, making $\exp(\tilde{x}B^\top C\tilde{x} - s) < \varepsilon'$, then $X^\top B^\top C\tilde{x} = \begin{bmatrix} W & b+s\mathbf{1} \\ 0 & s \end{bmatrix}\begin{bmatrix} x \\ 1 \end{bmatrix} = \begin{bmatrix} Wx + b + s\mathbf{1} \\ s \end{bmatrix}$. So we can compute a detailed form that is

$$\mathrm{T}^{\mathrm{softmax}}(x; X, Y)$$

$$= \begin{bmatrix} \dfrac{\sum\limits_{j=1}^{k} A_{1,j}\exp\left((Wx+b)_j + s\right)}{\sum\limits_{j=1}^{k}\exp\left((Wx+b)_j + s\right) + \exp\left(s\right) + \exp\left(\tilde{x}B^\top C\tilde{x}\right)} \\ \dfrac{\sum\limits_{j=1}^{k} A_{2,j}\exp\left((Wx+b)_j + s\right)}{\sum\limits_{j=1}^{k}\exp\left((Wx+b)_j + s\right) + \exp\left(s\right) + \exp\left(\tilde{x}B^\top C\tilde{x}\right)} \\ \vdots \\ \dfrac{\sum\limits_{j=1}^{k} A_{d_y,j}\exp\left((Wx+b)_j + s\right)}{\sum\limits_{j=1}^{k}\exp\left((Wx+b)_j + s\right) + \exp\left(s\right) + \exp\left(\tilde{x}B^\top C\tilde{x}\right)} \end{bmatrix} = \begin{bmatrix} \dfrac{\sum\limits_{j=1}^{k} A_{1,j}\exp\left((Wx+b)_j\right)}{\sum\limits_{j=1}^{k}\exp\left((Wx+b)_j\right) + 1 + \exp\left(\tilde{x}B^\top C\tilde{x} - s\right)} \\ \dfrac{\sum\limits_{j=1}^{k} A_{2,j}\exp\left((Wx+b)_j\right)}{\sum\limits_{j=1}^{k}\exp\left((Wx+b)_j\right) + 1 + \exp\left(\tilde{x}B^\top C\tilde{x} - s\right)} \\ \vdots \\ \dfrac{\sum\limits_{j=1}^{k} A_{d_y,j}\exp\left((Wx+b)_j\right)}{\sum\limits_{j=1}^{k}\exp\left((Wx+b)_j\right) + 1 + \exp\left(\tilde{x}B^\top C\tilde{x} - s\right)} \end{bmatrix}. \quad (44)$$

We focus on estimating the upper bound of the distence between $\mathrm{N}^{\mathrm{softmax}}$ and $\mathrm{T}^{\mathrm{softmax}}$, that is

$$\|\mathrm{N}^{\mathrm{softmax}}(x) - \mathrm{T}^{\mathrm{softmax}}(x; X, Y)\|$$

$$= \max_{1 \le i \le d_y}\left\{\left|\frac{\sum\limits_{j=1}^{k} A_{i,j}\exp\left((Wx+b)_j\right)}{\sum\limits_{j=1}^{k}\exp\left((Wx+b)_j\right) + 1} - \frac{\sum\limits_{j=1}^{k} A_{i,j}\exp\left((Wx+b)_j\right)}{\sum\limits_{j=1}^{k}\exp\left((Wx+b)_j\right) + 1 + \exp\left(\tilde{x}B^\top C\tilde{x} - s\right)}\right|\right\}$$

$$\le \|\mathrm{N}^{\mathrm{softmax}}\|\left|1 - \frac{\sum\limits_{j=1}^{k}\exp\left((Wx+b)_j\right) + 1}{\sum\limits_{j=1}^{k}\exp\left((Wx+b)_j\right) + 1 + \exp\left(\tilde{x}B^\top C\tilde{x} - s\right)}\right|$$

$$= \|\mathrm{N}^{\mathrm{softmax}}\|\left|\frac{\exp\left(\tilde{x}B^\top C\tilde{x} - s\right)}{\sum\limits_{j=1}^{k}\exp\left((Wx+b)_j\right) + 1 + \exp\left(\tilde{x}B^\top C\tilde{x} - s\right)}\right|$$

$$\le \|\mathrm{N}^{\mathrm{softmax}}\|\left|\exp\left(\tilde{x}B^\top C\tilde{x} - s\right)\right|$$

$$\le \|\mathrm{N}^{\mathrm{softmax}}\|\varepsilon'.$$

$$(45)$$

By setting $\varepsilon' = \frac{\varepsilon}{\|\mathrm{N}^{\mathrm{softmax}}(x)\|}$, which is ensured to be finite, the entire lemma has been proved. $\square$

## B    PROOF FOR SECTION 3

In this Appendix, we provide detailed proofs of the Proposition 4, Lemma 5, and Theorem 6 presented in Section 3.

### B.1    PROOF OF PROPOSITION 4

**Proposition 4.** *The scalar function $h_k(x) = \sum\limits_{i=1}^{k} a_i e^{b_i x}$, where $a_i, b_i, x \in \mathbb{R}$ and at least one $a_i$ is nonzero, has at most $k-1$ zero points.*

*Proof.* We prove this statement by induction. When $k = 1$ and 2, the statement is easy to prove. For the case $k = N$, suppose that every $h_N$ has at most $N - 1$ zero points.

Now consider $k = N + 1$. Let $h_{N+1}(x) = \sum_{i=1}^{N+1} a_i \exp(b_i x)$. Without loss of generality, assume $a_{N+1} \neq 0$. Thus, we can rewrite $h_{N+1}(x)$ as

$$h_{N+1}(x) = a_{N+1} e^{b_{N+1} x} \left( 1 + \sum_{i=1}^{N} \frac{a_i}{a_{N+1}} e^{(b_i - b_{N+1}) x} \right). \tag{46}$$

We proceed by contradiction. Suppose $h_{N+1}(x)$ has more than $N$ zero points. This implies

$$g(x) := 1 + \sum_{i=1}^{N} \frac{a_i}{a_{N+1}} e^{(b_i - b_{N+1}) x}, \tag{47}$$

has more than $N$ zero points.

Then, according to Rolle's Theorem, $g'(x)$ must have more than $N - 1$ zero points. Since $g'(x) = \sum_{i=1}^{N} \frac{a_i (b_i - b_{N+1})}{a_{N+1}} e^{(b_i - b_{N+1}) x}$ must have at least $N$ zero points, this leads to a contradiction.

Thus, $h_{N+1}(x) = \sum_{i=1}^{N+1} a_i e^{b_i x}$ can have at most $N$ zero points. The proof is complete. $\square$

### B.2 Proof of Lemma 5

**Lemma 5.** *The function class $\mathcal{N}_*^\sigma$, with elementwise or softmax activation $\sigma$, cannot achieve the UAP. Specifically, for any compact domain $\mathcal{K} \subset \mathbb{R}^{d_x}$, there exists a continuous function $f : \mathcal{K} \to \mathbb{R}^{d_y}$ and $\varepsilon_0 > 0$ such that*

$$\max_{x \in \mathcal{K}} \| f(x) - \mathrm{N}_*^\sigma(\tilde{x}) \| \geq \varepsilon_0, \quad \forall \, \mathrm{N}_*^\sigma \in \mathcal{N}_*^\sigma. \tag{48}$$

*Proof.* For any elementwise activations $\sigma$, the span of $\mathcal{N}_*^\sigma$, $\mathrm{span}(\mathcal{N}_*^\sigma)$, forms a finite-dimensional function space. $\mathcal{N}_*^\sigma$ is closed under the uniform norm supported by Theorem 1.21 from Rudin (1991) and Corollary C.4 from Cannarsa & D'Aprile (2015). This implies that the set of functions approximable by $\mathrm{span}(\mathcal{N}_*^\sigma)$ is precisely the set of functions within $\mathrm{span}(\mathcal{N}_*^\sigma)$. Consequently, any function not in $\mathrm{span}(\mathcal{N}_*^\sigma)$ cannot be arbitrarily approximated, meaning that the UAP cannot be achieved.

Without loss of generality, for any $\mathrm{N}_*^{\mathrm{softmax}} \in \mathcal{N}_*^{\mathrm{softmax}}$, assume $\mathcal{K} = [0, 1]^{d_x}$ and consider only the first component of $x$. Thus, we may assume $d_x = 1$. Let us consider the output of an arbitrary $j$-th dimension, that is

$$(\mathrm{N}_*^{\mathrm{softmax}})^{(j)} = \frac{\sum_{i=1}^{k} A_{j,i} \exp(w_i x + b_i)}{\sum_{l=1}^{k} \exp(w_l x + b_l)}. \tag{49}$$

Then the numerator, $\sum_{i=1}^{k} A_{j,i} \exp(w_i x + b_i)$, can have at most $k - 1$ zero points.

Now, we consider a special function $f(x) = \sin(mx)$, where $\lceil \frac{m}{\pi} \rceil > k - 1$, and the period is $T = \frac{2\pi}{m}$. $\lceil x \rceil$ is the smallest integer greater than or equal to $x$.

Let us focus on two adjacent extreme points $x_1, x_2$, where $f(x_1) = 1$ and $f(x_2) = -1$. We proceed by contradiction in our proof. Suppose $\mathcal{N}_*^{\mathrm{softmax}}$ can achieve the UAP. There exists $\mathrm{N}_*^{\mathrm{softmax}} \in \mathcal{N}_*^{\mathrm{softmax}}$ such that $|(\mathrm{N}_*^{\mathrm{softmax}})^{(j)} - f(x)| < \varepsilon$ for all $x \in [0, 1]$.

Taking $\varepsilon = 0.1$, we have:

$$|(\mathrm{N}_*^{\mathrm{softmax}}(x_1))^{(j)} - f(x_1)| < 0.1 \Rightarrow (\mathrm{N}_*^{\mathrm{softmax}}(x_1))^{(j)} > -0.1 + f(x_1) = 0.9,$$
$$|(\mathrm{N}_*^{\mathrm{softmax}}(x_2))^{(j)} - f(x_2)| < 0.1 \Rightarrow (\mathrm{N}_*^{\mathrm{softmax}}(x_2))^{(j)} < 0.1 + f(x_2) = -0.9,$$

By the intermediate value theorem, there exists some $x_0 \in (\min(x_1, x_2), \max(x_1, x_2))$, such that $(N_*^{\mathrm{softmax}}(x_0))^{(j)} = 0$ . Therefore, there is at least one zero of $(N_*^{\mathrm{softmax}}(x))^{(j)}$ between two adjacent extrema of $f(x)$, and the total number of zeros in the interval $[0, 1]$ is either $\lceil \frac{m}{\pi} \rceil + 1$ or $\lceil \frac{m}{\pi} \rceil$.

Thus, the number of zeros of $(N_*^{\mathrm{softmax}}(x))^{(j)}$ exceeds $k - 1$, leading to a contradiction.

If approximation cannot be achieved in one dimension, it is evident that it cannot be achieved in higher dimensions either. Therefore, $\mathcal{N}_*^{\mathrm{softmax}}$ cannot achieve the UAP. $\qquad\square$

### B.3 PROOF OF THEOREM 6

**Theorem 6.** *The function class $\mathcal{T}_*^\sigma$, with elementwise or softmax activation $\sigma$, cannot achieve the UAP. Specifically, for any compact domain $\mathcal{K} \subset \mathbb{R}^{d_x - 1}$, there exists a continuous function $f : \mathcal{K} \to \mathbb{R}^{d_y}$ and $\varepsilon_0 > 0$ such that*

$$\max_{x \in \mathcal{K}} \| f(x) - T_*^\sigma(\tilde{x}) \| \geq \varepsilon_0, \quad \forall\, T_*^\sigma \in \mathcal{T}_*^\sigma. \tag{50}$$

*Proof.* For any $T_*^\sigma \in \mathcal{T}_*^\sigma$ with elementwise activation $\sigma$, since $T^\sigma = N^\sigma$, we can replace $N_*^\sigma$ in Lemma 5 with $T_*^\sigma$ accordingly.

Without loss of generality, for any $T_*^{\mathrm{softmax}} \in \mathcal{T}_*^{\mathrm{softmax}}$, assume $\mathcal{K} = [0, 1]^{d_x}$ and consider the output of an arbitrary $j$-th dimension and one-dimensional input as an example that is

$$(T_*^{\mathrm{softmax}})^{(j)} = \frac{\sum\limits_{i=1}^{k} A_{j,i} \exp\left(w_i x + b_i\right)}{\sum\limits_{i=1}^{k} \exp\left(w_i x + b_i\right) + \exp\left(\tilde{x}^\top B^\top C^\top \tilde{x}\right)}. \tag{51}$$

We observe that the form of the numerator remains consistent with Lemma 5, and we follow the same proof as above. We consider a specific function $f(x) = \sin(mx)$, where $\lceil \frac{m}{\pi} \rceil > k - 1$, and its period is $T = \frac{2\pi}{m}$. This leads to the conclusion that $\mathcal{T}_*^{\mathrm{softmax}}$ cannot achieve the UAP. $\qquad\square$

## C  PROOF FOR SECTION 4

In this Appendix, we introduce Lemma 12 to assist in the proof of Theorem 8 and utilize Lemma 13 to provide a detailed proof of Theorem 9.

### C.1 PROOF OF LEMMA 12

**Lemma 12.** *For a network with a fixed width and a continuous activation function, it is possible to apply slight perturbations within an arbitrarily small error margin. For any network $N_1^\sigma(x)$ defined on a compact set $\mathcal{K} \subset \mathbb{R}^{d_x}$, with parameters $A \in \mathbb{R}^{d_y \times k}, W \in \mathbb{R}^{k \times d_x}, b \in \mathbb{R}^{k \times 1}$, there exists $M > 0 (\|x\| < M)$, and for any $\varepsilon > 0$, there exists $0 < \delta < \frac{\varepsilon}{k}$ and a perturbed network $N_2^\sigma(x)$ with parameters $\tilde{A} \in \mathbb{R}^{d_y \times k}, \tilde{W} \in \mathbb{R}^{k \times d_x}, \tilde{b} \in \mathbb{R}^{k \times 1}$, such that if $\max\{\|a_i - \tilde{a}_i\|, M\|w_i - \tilde{w}_i\| + \|b - \tilde{b}\| \mid i = 1, \cdots, k\} < \delta$, then*

$$\| N_1(x) - N_2(x) \| < \varepsilon^2, \quad \forall x \in \mathcal{K}, \tag{52}$$

*where $a_i, \tilde{a}_i$ are the $i$-th column vectors of $A, \tilde{A}$, respectively, $w_i, \tilde{w}_i$ are the $i$-th row vectors of $W, \tilde{W}$, and $b_i, \tilde{b}_i$ are the $i$-th components of $b, \tilde{b}$, respectively, for any $i = 1, \cdots, k$.*

*Proof.* We have $N_1^\sigma(x) = \sum\limits_{i=1}^{k} a_i \sigma(w_i x + b_i)$, where $a_i \in \mathbb{R}^{d_y}, w_i \in \mathbb{R}^{d_x}, b_i \in \mathbb{R}$, and $\tilde{N}_2^\sigma(x) = \sum\limits_{i=1}^{k} \tilde{a}_i \sigma(\tilde{w}_i x + \tilde{b}_i)$, where $\tilde{a}_j \in \mathbb{R}^{d_y}, \tilde{w}_i \in \mathbb{R}^{d_x}, \tilde{b}_i \in \mathbb{R}$. For any $x \in \mathcal{K}, \|x\| < M$.

Due to the continuity of the activation function, for any $\varepsilon > 0$, there exists $0 < \delta < \frac{\varepsilon}{k}$ such that if $\|w_i x + b_i - (\tilde{w}_i x + \tilde{b}_i)\| \leq \|w_i - \tilde{w}_i\|\|x\| + \|b_i - \tilde{b}_i\| < M\|w_i - \tilde{w}_i\| + \|b - \tilde{b}\| < \delta, i = 1, \cdots, k,$ then $\|\sigma(w_i x + b_i) - \sigma(\tilde{w}_i x + \tilde{b}_i)\| < \varepsilon, i = 1, \cdots, k,$ and $\|a_i - \tilde{a}_i\| < \delta, i = 1, \cdots, k.$

Combining all these inequalities, we can further derive:

$$
\begin{aligned}
\|\mathrm{N}_1^\sigma(x) - \mathrm{N}_2^\sigma(x)\| &= \|\sum_{i=1}^k a_i \sigma(w_i x + b_i) - \sum_{i=1}^k \tilde{a}_i \sigma(\tilde{w}_i x + \tilde{b}_i)\| \\
&\leq k \max\{\|a_i - \tilde{a}_i\| \mid i = 1, \cdots, k\} \max\{\|\sigma(w_i x + b_i) - \sigma(\tilde{w}_i x + \tilde{b}_i)\| \mid i = 1, \cdots, k\} \\
&< \varepsilon^2
\end{aligned}
\tag{53}
$$

The proof is complete. $\qquad\square$

## C.2 PROOF OF THEOREM 8

**Theorem 8.** *Let $\mathcal{T}_{*,\mathcal{P}}^\sigma$ be the class of functions $\mathrm{T}_{*,\mathcal{P}}^\sigma$, where $\sigma$ is an elementwise activation, the subscript refers the finite vocabulary $\mathcal{V} = \mathcal{V}_x \times \mathcal{V}_y$, $\mathcal{P} = \mathcal{P}_x \times \mathcal{P}_y$ represents the positional encoding map, and denote the set $S$ as:*

$$
S := \mathcal{V}_x + \mathcal{P}_x = \left\{ x_i + \mathcal{P}_x^{(j)} \mid x_i \in \mathcal{V}_x, i, j \in \mathbb{N}^+ \right\}. \tag{54}
$$

*If $S$ is dense in $\mathbb{R}^{d_x}$, $\{1, -1, \sqrt{2}, 0\}^{d_y} \subset \mathcal{V}_y$ and $\mathcal{P}_y = 0$, then $\mathcal{T}_{*,\mathcal{P}}^\sigma$ can achieve the UAP. That is, for any continuous function $f : \mathbb{R}^{d_x-1} \to \mathbb{R}^{d_y}$ defined on a compact domain $\mathcal{K}$, and for any $\varepsilon > 0$, there always exist $X \in \mathbb{R}^{d_x \times n}$ and $Y \in \mathbb{R}^{d_y \times n}$ from the vocabulary $\mathcal{V}$ (i.e., $x^{(i)} \in \mathcal{V}_x, y^{(i)} \in \mathcal{V}_y$) with some length $n \in \mathbb{N}^+$ such that*

$$
\left\| \mathrm{T}_{*,\mathcal{P}}^\sigma (\tilde{x}; X, Y) - f(x) \right\| < \varepsilon, \quad \forall x \in \mathcal{K}. \tag{55}
$$

*Proof.* Our conclusion holds for all element-wise continuous activation functions in $\mathcal{T}_{*,\mathcal{P}}^\sigma$. We demonstrate this with $d_y = 1$. Similar cases can be inferred by analogy.

We reformulating the problem.

Using Lemma 2, we have,

$$
\mathrm{T}_{*,\mathcal{P}}^\sigma (\tilde{x}; X, Y) = UY_\mathcal{P} \, \sigma \left( (X + \mathcal{P})^\top B^\top C \tilde{x} \right) = UY_\mathcal{P} \, \sigma \left( X_\mathcal{P}^\top B^\top C \tilde{x} \right). \tag{56}
$$

Since $\mathcal{P}_y = 0$, it follows that $Y_\mathcal{P} = Y$. For any continuous function $f : \mathbb{R}^{d_x-1} \to \mathbb{R}^{d_y}$ defined on a compact domain $\mathcal{K}$ and for any $\varepsilon > 0$, we aim to show that there exists $\mathrm{T}_{*,\mathcal{P}}^\sigma \in \mathcal{T}_{*,\mathcal{P}}^\sigma$ such that:

$$
\begin{aligned}
&\left\| \mathrm{T}_{*,\mathcal{P}}^\sigma \left( \begin{bmatrix} x \\ 1 \end{bmatrix}; X, Y \right) - Uf(x) \right\| < \|U\|\varepsilon, \quad \forall x \in \mathcal{K}, \\
&\Leftrightarrow \left\| Y \, \sigma \left( X_\mathcal{P}^\top B^\top C \tilde{x} \right) - f(x) \right\| < \varepsilon, \quad \forall x \in \mathcal{K}.
\end{aligned}
\tag{57}
$$

Let $\mathrm{N}_*^\sigma(x) := Y \, \sigma \left( X_\mathcal{P}^\top B^\top C \tilde{x} \right) = \sum_{i=1}^n y^{(i)} \, \sigma(\tilde{R}_i \tilde{x}) \in \mathcal{N}_*^\sigma$, where $n \in \mathbb{N}^+$, $y^{(i)} \in \mathbb{R}^{d_y}$ and $\tilde{R}_i \in \mathbb{R}^{d_x}$ (the $i$-th row of $\tilde{R} \in \mathbb{R}^{n \times d_x}$). The proof is divided into four steps:

**Step (1)**: Approximating $f(x)$ Using a $\mathrm{N}^\sigma(x)$

For any $\varepsilon > 0$, there exists a neural network $\mathrm{N}^\sigma(x) = A \, \sigma(Wx + b) = \sum_{i=1}^k a_i \, \sigma(w_i x + b_i) \in \mathcal{N}^\sigma$, with parameters $k \in \mathbb{N}^+$, $A \in \mathbb{R}^{d_y \times k}$, $b \in \mathbb{R}^k$, and $W \in \mathbb{R}^{k \times (d_x-1)}$ (where $a_i$ and $w_i$ denote the $i$-th column of $A$ and the $i$-th row of $W$),

$$
\|A \, \sigma(Wx + b) - f(x)\| < \frac{\varepsilon}{3}, \quad \forall x \in \mathcal{K}, \tag{58}
$$

which is supported by Lemma 1.

**Step (2)**: Approximating $\mathrm{N}^\sigma(x)$ Using $\mathrm{N}'(x)$

Using Lemma 7 and Lemma 12, a neural network $\mathrm{N}^\sigma(x) = \sum_{i=1}^{k} a_i \sigma(w_i x + b_i) \in \mathcal{N}^\sigma$ can be perturbed into $\mathrm{N}'(x) = \sum_{i=1}^{k} (q\sqrt{2} \pm l)_i \, \sigma(\tilde{w}_i x + \tilde{b}_i)$ (with $q_i \in \mathbb{N}^+$ and $l_i \in \mathbb{N}^+, i = 1, \cdots, k$), such that for any $\varepsilon > 0$, there exists $0 < \delta < \frac{\varepsilon}{k}$ satisfying:

$$\max\{\|a_i - (q\sqrt{2} \pm l)_i\|_{\max}, M\|w_i - \tilde{w}_i\|_{\max} + \|b - \tilde{b}\|_{\max} \mid i = 1, \cdots, k\} < \delta, \qquad (59)$$

ensuring:

$$\|\mathrm{N}^\sigma(x) - \mathrm{N}'(x)\| = \left\| \sum_{i=1}^{k} a_i \, \sigma(w_i x + b_i) - \sum_{i=1}^{k} (q\sqrt{2} \pm l)_i \, \sigma(\tilde{w}_i x + \tilde{b}_i) \right\| < \frac{\varepsilon}{3}, \quad \forall x \in \mathcal{K}. \quad (60)$$

**Step (3)**: Approximating $\mathrm{N}'(x)$ Using $\mathrm{N}_*^\sigma(x)$

Next, we show that $\mathrm{N}_*^\sigma(x) = \sum_{i=1}^{n} y^{(i)} \sigma(\tilde{R}_i \tilde{x}) \in \mathcal{N}_*^\sigma$ can approximate $\mathrm{N}'(x) = \sum_{i=1}^{k} (q\sqrt{2} \pm l)_i \, \sigma(\tilde{w}_i \tilde{x})$. As a demonstration, we approximate a single term $(q\sqrt{2} \pm l)_1 \, \sigma(\tilde{w}_1 \tilde{x})$.

Given that the set $S$ is dense in $\mathbb{R}^{d_x}$, it follows that $G := \{\tilde{R} \mid \tilde{R} = X_{\mathcal{P}}^\top B^\top C, X_{\mathcal{P}} \subset 2^S\}$ is also dense. Since $y^{(i)} \in \{1, -1, \sqrt{2}, 0\}$, we require $q_1 + l_1$ elements of $\tilde{R}_i$ to approximate $\tilde{w}_1$ such that

$$\left\| \sum_{j \in K_1} y^{(j)} \sigma(\tilde{R}_j \tilde{x}) - (q\sqrt{2} \pm l)_1 \, \sigma(\tilde{w}_1 \tilde{x}) \right\|$$
$$= \|\sqrt{2} \sum_{j \in Q_1} \sigma(\tilde{R}_j \tilde{x}) \pm \sum_{j \in L_1} \sigma(\tilde{R}_j \tilde{x}) - (q\sqrt{2} \pm l)_1 \, \sigma(\tilde{w}_1 \tilde{x})\| \qquad (61)$$
$$< \frac{\varepsilon}{3k}, \quad \forall x \in \mathcal{K}.$$

Here, $\#(K_1) = q_1 + l_1$ and $K_1 = Q_1 \bigcup L_1$, where $Q_1, L_1$ are disjoint subsets of positive integer indices satisfying $\#(Q_1) = q_1$ and $\#(L_1) = l_1$. For this construction, we assign $y^{(j)} = \sqrt{2}$ for $j \in Q_1$ and $y^{(j)} = \pm 1$ for $j \in L_1$. For $j \notin \bigcup_{l=1}^{k} K_l$, we set $y^{(j)} = 0$. We then define $n = \max\{j \mid j \in \bigcup_{l=1}^{k} K_l\}$.

Finally, we have:

$$\left\| \mathrm{N}_*^\sigma(x) - \mathrm{N}'(x) \right\| = \| \sum_{i=1}^{n} y^{(i)} \sigma(\tilde{R}_i \tilde{x}) - \sum_{i=1}^{k} (q\sqrt{2} \pm l)_i \sigma(\tilde{w}_i \tilde{x}) \| < \frac{\varepsilon}{3}, \quad \forall x \in \mathcal{K}.$$

**Step (4)**: Combining Results

Combining all results, we have:

$$\|Y \sigma \left( X_{\mathcal{P}}^\top B^\top C \tilde{x} \right) - f(x)\| = \|\mathrm{N}_*^\sigma(x) - f(x)\|$$
$$< \|\mathrm{N}_*^\sigma(x) - \mathrm{N}'(x)\| + \|\mathrm{N}'(x) - \mathrm{N}^\sigma(x)\| + \|\mathrm{N}^\sigma(x) - f(x)\| \quad (62)$$
$$< \varepsilon, \quad \forall x \in \mathcal{K}.$$

The proof is complete.

$\square$

## C.3 Proof of Theorem 9

**Lemma 13.** *For any continuous function $f : \mathbb{R}^{d_x} \to \mathbb{R}^{d_y}$ defined on a compact domain $\mathcal{K}$ and $\varepsilon > 0$, there always exist a softmax FNN $\mathrm{N}^{\exp}(x) : \mathbb{R}^{dx} \to \mathbb{R}^{d_y}, x \mapsto A \exp(Wx + b)$ satisfying*

$$\|\mathrm{N}^{\exp}(x) - f(x)\| < \varepsilon, \quad \forall x \in \mathcal{K}$$

*where $b = 0$ and all row vector of $W$ are restricted in a neighborhood $B(w^*, \delta)$ with any prefixed $w^* \in \mathbb{R}^{d_x}$ and $\delta > 0$.*

*Proof.* According to Stone-Weierstrass theorem we know that, for any continuous function $f$ and $i = 1, \cdots, d_y$ and $\varepsilon' > 0$, thers exists a polynomial $P_i(x)$ which can approximate $\exp(-w^* \cdot x)(f(x))_i$, i.e.

$$\|P_i(x) - \exp(-w^* \cdot x)(f(x))_i\| < \varepsilon', \quad \forall x \in \mathcal{K}. \tag{63}$$

The inequation above indicates that

$$\|\exp(w^* \cdot x)P_i(x) - (f(x))_i\| < \|\exp(w^* \cdot x)\|\varepsilon' := \varepsilon, \quad \forall x \in \mathcal{K}. \tag{64}$$

Then we construct a FNN with exponential activation function to approximate $\exp(w^* \cdot x)P_i(x)$. Without loss of generality, let us consider the first hidden neuron of a softmax FNN. Assume

$$h(w) = \exp(w \cdot x) = \exp(w_1 x_1 + \cdots + w_{d_x} x_{d_x}), \tag{65}$$

then the multiple derivatives of $h(w)$ with respect to $w_1, \cdots, w_{d_x}$ is

$$\frac{\partial^{|\alpha|} h}{\partial w^\alpha} = \frac{\partial^\alpha h}{\partial w_1^{\alpha_1} \cdots \partial w_{d_x}^{\alpha_{d_x}}} \tag{66}$$

where $\alpha \in \mathbb{N}^{d_x}$ represents the index and $|\alpha| := \alpha_1 + \cdots \alpha_{d_x}$. Actually, the form of $\frac{\partial^\alpha h}{\partial w_1^{\alpha_1} \cdots \partial w_{d_x}^{\alpha_{d_x}}}$ is a polynomial of $|\alpha|$ degree with respect to $x_1, \cdots, x_k$ times $h(w)$. Note that $\exp(w^* \cdot x)P_i(x)$ can be written as a finite sum of some multiple derivatives of $h(x)$, that is

$$\exp(w^* \cdot x)P_i(x) = \left( \sum_{\alpha \in \Lambda_i} a_\alpha \frac{\partial^{|\alpha|} h}{\partial w^\alpha} \right)\bigg|_{w=w^*}, \tag{67}$$

where $\alpha \in \mathbb{N}^{d_x}$ is the index of multiple derivative and $\Lambda_i$ is a finite multiple set of indexes. As for multiple derivatives, they can be approximated by finite difference method, and the approach of finite difference method can be done by a one hidden layer. For example,

$$\begin{aligned}
x_1 \exp(w^* \cdot x) &= \frac{\partial h}{\partial w_1}\bigg|_{w=w^*} \\
&= \frac{h(w^* + \lambda e_1) - h(w^*)}{\lambda} + R_1(\lambda, w^*) \\
&= \frac{1}{\lambda} \exp((w^* + \lambda e_1) \cdot x) - \frac{1}{\lambda} \exp(w^* \cdot x) + R_1(\lambda, w^*),
\end{aligned} \tag{68}$$

and

$$\begin{aligned}
x_1 x_2 \exp(w^* \cdot x) &= \frac{\partial^2 h}{\partial w_1 \partial w_2}\bigg|_{w=w^*} \\
&= \frac{h(w^* + \lambda(e_1 + e_2)) - h(w^* + \lambda e_1) - h(w^* + \lambda e_2) + h(w^*)}{\lambda^2} + R_2(\lambda, w^*) \\
&= \frac{1}{\lambda^2} \exp((w^* + \lambda(e_1 + e_2)) \cdot x) - \frac{1}{\lambda^2} \exp((w^* + \lambda e_1) \cdot x) - \\
&\quad \frac{1}{\lambda^2} \exp((w^* + \lambda e_2) \cdot x) + \frac{1}{\lambda^2} \exp(w^* \cdot x) + R_2(\lambda, w^*),
\end{aligned} \tag{69}$$

where $e_1 = (1, 0, 0, \cdots, 0), e_2 = (0, 1, 0, \cdots, 0)$ are unit vectors and $R_1(\lambda, w^*)$ and $R_2(\lambda, w^*)$ are error terms with respect to $\lambda$ and $w^*$. The error term $R_1(\lambda, w^*) = \lambda \frac{\partial^2 h}{\partial w_1^2}\big|_{w=\xi}$ for some $\xi$ between $w^*$ and $w^* + \lambda e_1$. It is obvious that the partial differential term is bounded in $B(w^*, \delta)$, so the error can be controlled by $\lambda$. For $R_2(\lambda, w^*)$ it is similar. Equation (69) holds, as shown in Chapter X of Boole (2009).

Since $\lambda$ is very small and the exponential terms $\exp(w^* \cdot x)$ only involve the parameters $w^*$, $w^* + \lambda e_1$ and $w^* + \lambda e_2$, which all lie within a small neighborhood of $w^*$ the desired conclusion can be drawn, and this means we can actually restrict that all row vectors of $W$ are restricted in the neighborhood $B(w^*, \delta)$. $\qquad\square$

**Theorem 9.** *If the set $S$ is dense in $[-1,1]^{d_x}$, then $\mathcal{T}_{*,\mathcal{P}}^{\mathrm{ReLU}}$ is capable of achieving the UAP. Additionally, if $S$ is only dense in a neighborhood $B(w^*, \delta)$ of a point $w^* \in \mathbb{R}^{d_x}$ with radius $\delta > 0$, then the class of transformers with exponential activation, $\mathcal{T}_{*,\mathcal{P}}^{\mathrm{exp}}$, is capable of achieving the UAP.*

*Proof.* For the proof of ReLU case, we follow the same reasoning as in the previous one, noting that $\mathrm{ReLU}(ax) = a\,\mathrm{ReLU}(x)$ holds for any positive $a$. In the proof of Theorem 8, we construct a $\mathrm{T}_{*,\mathcal{P}}^{\mathrm{ReLU}}$ to approximate a FNN $A\,\mathrm{ReLU}(Wx + b)$. Here we can do the similar construction to find another $\tilde{\mathrm{T}}_{*,\mathcal{P}}^{\mathrm{ReLU}}$ to approximate $tA\,\mathrm{ReLU}\left(\frac{W}{t}x + b\right)$ as the second to the forth steps in Theorem 8, where $t$ is big enough to make the elements in $W$ is small enough so $S = \{x_i + \mathcal{P}_x^{(j)} \mid x_i \in \mathcal{V}_x, i, j \in \mathbb{N}^+\}$ is dense in $[-1,1]^{d_x}$ is sufficient. For the exponential, we using Lemma 13, we can do step the second to the forth steps in Theorem 8 again, which is similar to ReLU case. $\square$

## D  GENERAL CASE FOR THEOREM 6

It is important to note that Theorem 6 remains valid even without imposing specific constraints on the value, query, and key matrices $V$, $Q$, and $K$ (e.g., the sparse partition described in equation (14)). Below, we outline the reasoning.

In general, we decompose the matrices as follows:

$$Q^\top K = \begin{bmatrix} M_{11} & M_{12} \\ M_{21} & M_{22} \end{bmatrix}, V = \begin{bmatrix} D & E \\ F & U \end{bmatrix}, \tag{70}$$

where $M_{11}, D \in \mathbb{R}^{d_x \times d_x}$, $M_{12}, E \in \mathbb{R}^{d_x \times d_y}$, $M_{21}, F \in \mathbb{R}^{d_y \times d_x}$, and $M_{22}, U \in \mathbb{R}^{d_y \times d_y}$, respectively.

The attention mechanism can then be computed as:

$$\begin{aligned}
\mathrm{Attn}_{Q,K,V}^\sigma(Z) &= VZM\sigma(Z^\top Q^\top KZ) \\
&= \begin{bmatrix} D & E \\ F & U \end{bmatrix} \begin{bmatrix} X & x \\ Y & 0 \end{bmatrix} \begin{bmatrix} I_n & \\ & 0 \end{bmatrix} \sigma\left( \begin{bmatrix} X^\top & Y^\top \\ x^\top & 0 \end{bmatrix} \begin{bmatrix} M_{11} & M_{12} \\ M_{21} & M_{22} \end{bmatrix} \begin{bmatrix} X & x \\ Y & 0 \end{bmatrix} \right) \\
&= \begin{bmatrix} DX + EY & 0 \\ FX + UY & 0 \end{bmatrix} \sigma\left( \begin{bmatrix} M & (X^\top M_{11} + Y^\top M_{21})x \\ x^\top(M_{11}X + W_{12}Y) & x^\top M_{11}x \end{bmatrix} \right),
\end{aligned}$$

where $M$ represents the matrix $X^\top M_{11}X + X^\top W_{12}Y + Y^\top M_{21}X + Y^\top M_{22}Y$. As a result, we have:

$$\mathrm{T}^\sigma(\tilde{x}; X, Y) = (FX + UY)\sigma((X^\top M_{11} + Y^\top M_{21})\tilde{x}), \tag{71}$$

for the case of elementwise activations, and:

$$\mathrm{T}^{\mathrm{softmax}}(\tilde{x}; X, Y) = (FX + UY)\left[\mathrm{softmax}\left(\begin{bmatrix} (X^\top M_{11} + Y^\top M_{21})\tilde{x} \\ \tilde{x}^\top M_{11}\tilde{x} \end{bmatrix}\right)\right]_{1:n}, \tag{72}$$

for the case of softmax activation.

By revisiting the definition of $\mathrm{T}_*^\sigma(x; X, Y)$ and comparing $\mathrm{T}_*^\sigma$ and $\mathrm{T}_*^{\mathrm{softmax}}$ presented here with those in Appendix B, it is clear that the only distinction lies in the specific matrices involved. Consequently, the proof process for Theorem 6 can be directly applied to obtain the same results.

