# OpenReview forum: "Vocabulary In-Context Learning in Transformers: Benefits of Positional Encoding"
_ICLR.cc/2025/Conference — Submitted to ICLR 2025_

### Official Review · Reviewer_79Bt · 2024-10-26

**Soundness:** 3
**Presentation:** 3
**Contribution:** 3
**Rating:** 6
**Confidence:** 3

**Summary:**

The most important contribution of this work is theoretic: In (single-layer) Transformer,s (vocabulary) in-context learning can only (possibly) achieve UAP when positional encoding is there. I am not an expert in this field, but compared to the most known Transformer ICL+UAP research to my knowledge (e.g., Yun et al., Luo et al., Petrov et al.), this work emphasizes the positional encoding in Transformer.

**Strengths:**

- The theoretical insight and contribution: The promise of this work is intriguing to me, as it’s the first time I’ve seen research attempting to bridge the theoretical understanding of in-context learning from positional encoding through the lens of universal adversarial perturbations.
- The proofs (especially Theorem 6). With a finite vocabulary and no positional encoding, the authors prove that single-layer Transformers cannot achieve universal approximation properties (UAP) for ICL tasks.
- The discussion of ReLU is a plus to me.

**Weaknesses:**

This paper makes a good number of assumptions.
- I accept most of the assumptions made as valid, but...
- I would prefer the authors mention all assumptions more clearly in bullets or “Assumption $n$” like the Theorems. For example, Lines 378-379 state an important assumption of the density property using Diophantine approximation. I almost missed it…
- The part I am a bit concerned about is that this work only studies absolute positional encodings and single-layer transformers. The latter is fine, but I think the authors should discuss (at least on a higher level) how the results of this work could potentially generalize to other PEs like RPE and RoPE.

[Minor] In Sec 1.2 where positional encodings are discussed, I believe that the Rotary Position Embedding (RoPE) [1] should be mentioned.

[1] Su, Jianlin, et al. "Roformer: Enhanced transformer with rotary position embedding." Neurocomputing 568 (2024): 127063.

**Questions:**

**Question 1**: In line 160 “Unlike the setting in Ahn et al. (2024); Cheng et al. (2024), in this paper, we do not assume a correspondence between $x^{(i)}$ and $y^{(i)}$” Are you suggesting that there is **independence** between  $x^{(i)}$  and $y^{(i)}$, or that they are **unpaired data** in a **weakly supervised setting**? Or do you mean there may be **false** or **unmatched pairs** of $x$ and $y$?

**Question 2**: Can the results generalize to RPE and RoPE? Could authors include a discussion section on how their results might extend to or differ for other types of positional encodings like RPE and RoPE.

---

> ### Author Response · Authors · 2024-11-24
> **Author's responses**
>
> Thank you for your thorough review and valuable comments. We have carefully considered your suggestions, and our detailed responses are provided below:
>
> - **Assumptions.** We appreciate your suggestion. To enhance clarity and emphasis, we have revised the manuscript by presenting the key assumptions within a dedicated *Assumption* environment (page 5).
>
> - **Statement in Lines 378–379.** Thank you for highlighting this issue. This statement is not an assumption but rather a mathematical result derived from Lemma 7. We have updated the manuscript to clarify this distinction.
>
> - **Correspondence in Line 160.** Thank you for your careful observation. The intended meaning here is that the data are unpaired, implying independence between $x^{(i)}$ and $y^{(i)}$. In contrast to the settings in Ahn et al. (2024) and Cheng et al. (2024), where $y^{(i)}$ is determined as $\phi(x^{(i)})$, our framework assumes that $x^{(i)}$ and $y^{(i)}$ are independently sampled from $\mathcal{V}_x$ and $\mathcal{V}_y$, respectively. We have revised the manuscript to make this distinction clearer.
>
> - **Beyond Absolute Positional Encodings.** Thank you for this suggestion. We have focused on the case of absolute positional encodings (APE) because it offers a more straightforward framework for analysis. Relative positional encodings (RPE) and rotary positional encodings (RoPE) present additional complexities due to their varied implementations. We have revised the manuscript to highlight this rationale and identified these cases as potential directions for future work. Additionally, we have included relevant references on positional encodings to enrich the discussion.

---

> > ### Comment · Reviewer_79Bt · 2024-11-26
> >
> > Equation (70) introduces an error term without clearly bounding or explaining its significance. I am confused by what authors mean by "due to the approach of finite difference method, we can actually restrict the weight in a small neighborhood", and why this indicates "that all row vectors of W are restricted in its neighborhood"

---

> > > ### Author Response · Authors · 2024-11-26
> > > **Approach of finite difference method**
> > >
> > > Thank you for your question.
> > >
> > > Finite differences are a fundamental method in numerical analysis. The error term here is a higher-order infinitesimal of the step size $\lambda$, specifically $R(\lambda, w^*) = o(\lambda)$. When $\lambda$ is sufficiently small, equation (32) closely approximates the corresponding derivative. The same conclusion holds for higher-order derivatives.
> > >
> > > Since $\lambda$ is very small and the exponential terms in equation (32) only involve the parameters $w^*$ and $w^* + \lambda e_1$, which both lie within a small neighborhood of $w^*$, the desired conclusion can be drawn. Given that the rows of $W$ are constructed using the same approach, they also lie within a small neighborhood of $w^*$.
> > >
> > > We will include more details in the next revision of the manuscript to make it more self-contained and accessible to readers who may not be familiar with finite difference methods.

---

> > > > ### Comment · Reviewer_79Bt · 2024-11-28
> > > >
> > > > Thank you for your response. My concerns are addressed.

---

### Official Review · Reviewer_QJXB · 2024-11-03

**Soundness:** 3
**Presentation:** 4
**Contribution:** 3
**Rating:** 8
**Confidence:** 2

**Summary:**

The paper provides a mathematical theory of how adding positional encoding information to input tokens affects a transformer's ability to exhibit the Universal Approximation Property (UAP).
This is the study of when a simple, single-layer transformer can function as a universal approximator - that is, it can model any continuous function with arbitrary precision. The researchers showed that positional coding plays a crucial role: without it, single-layer transformers fall short of universal approximation.
A key aspect of the mathematical proof is the use of the Kronecker Approximation Theorem to establish conditions that ensure the density of the sum of the positional encoding and the input tokens. This theorem allows us to show that positional encoding provides a sufficiently rich sum of tokens.
 This result underscores that positional encoding is not only helpful, but essential for transformers to effectively model complex patterns, especially when working with limited, discrete vocabularies, from a UAP perspective.

**Strengths:**

-The paper advances the theoretical understanding of in-context learning and its relation to positional encoding in transformers, especially under finite vocabulary constraints, by mathematically proving that positional encoding allows transformers to achieve UAP.
- The mathematical proof is based on the effective f use of the Kronecker Approximation Theorem, and the rational formulation of the theory.
- The paper provides clear conditions for UAP: The study defines explicit mathematical conditions and sufficient criteria for positional encoding to enable UAP, providing a basis for further theoretical and applied research in NLP.

**Weaknesses:**

- Although the theoretical contribution is clear, the practical insights for practitioners are limited.
- This paper lacks experimental validation. While the mathematical differences are qualitatively clear, there are no experimental demonstrations to show how these theoretical differences translate to actual differences in model performance.

**Questions:**

- Is it possible to provide some numerical demonstration that clearly shows that the difference suggested by this theory in positional encoding actually produces different performance?
- In the discussion, it may be better to describe a bit more about the theoretical insights based on the understanding gained through the proof for practitioners who will be designing positional encoding or just using Transformer.

---

> ### Author Response · Authors · 2024-11-24
> **Author's responses**
>
> Thank you for your positive comments and valuable suggestions.
>
> While we would like to further strengthen the conclusions of this paper through additional experiments, this is currently beyond the scope of our expertise. Instead, we have added several sentences in the revised manuscript to offer potential insights for practitioners. Specifically, in Remark 10, we note that certain algorithms use function composition to enumerate numbers that are dense in $\mathbb{R}$. This idea could be extended to design positional encodings by composing fixed functions, similar to the approach employed in RNNs. RNNs integrate information from each position in a sequence, thereby capturing the importance of word order in sentence meaning. However, to the best of our knowledge, existing research on RNNs has not explored the denseness properties of the sets formed by their hidden state sequences. We hope this property will inspire experimental researchers in future studies.

---

> > ### Comment · Reviewer_QJXB · 2024-11-27
> >
> > Thank you for your reply.
> > After reading your comments, I will not change my rating.

---

### Official Review · Reviewer_Tz2M · 2024-11-04

**Soundness:** 3
**Presentation:** 2
**Contribution:** 2
**Rating:** 5
**Confidence:** 2

**Summary:**

This paper is about the approximation properties of transformers in ICL: The authors fix the transformer
weights V ,Q,K, then given a target function f, they aim to adjust the content of the context so that the output of the Transformer network can approximate f. The claim is that when there is no restriction on the size of the vocabulary, single layer transformers have the universal approximation property in ICL. But when the size of the vocabulary is limited, single layer transformers do not have the
universal approximation property in ICL, which is remedied by allowing for position encodings.

**Strengths:**

ICL is poorly understood, and this paper makes a step towards understanding the capacity that transformers have for ICL and the role of positional encodings.

**Weaknesses:**

A lot of the paper is spent setting up notation. However, many statements are imprecise and opaque. For example, the proof sketch of theorem 8 is quite hard to follow.

Logically, the idea of fixing the network weights and "adjusting the content of the context so that the output of the Transformer network can approximate f" doesn't commonly arise in practical scenarios. For example, Lemma 3 finds a vocabulary matrix X, Y to fit an arbitrary function f. But after the model is trained, in a typical ICL setting the model has to learn to adapt to the arbitrary function f on the fly given the fixed vocabulary matrix and weights V, Q, K. That is, one does not optimize over the context to find the function.

No experiments or simulations to illustrate the result -- not a serious problem for a theory paper but does take away the significance of this work.

**Questions:**

Theorem 8: However, in natural tasks, we don't have a choice over the content of the context, we are
given sequences and yet the network must fit the target function f in-context. Could the authors please elaborate on why this is a reasonable setting?

Line 398: "From previous work" -- which one?

Is there a missing hypothesis in line 383/theorem 8: "If S = \{x_i + P^{(j)}_x ∣ x_i \in Vx, i \in N_+\} is
dense in R^{dx} , and"

Line 409 "The contradiction arises from the finite nature of the vocabulary, which limits the finiteness of UY and X⊤B⊤C". What does finiteness mean here?

Line 410 "We invoke the density of the set S = {xi + P(j) x ∣ xi ∈ Vr, j ∈ N+} in Rdx ,
which ensures the density of X⊤T B⊤C." But don't we learn a particular position encoding which is then fixed for the
transformer, so for a particular transformer S can't be dense -- doesn't this defeat their paper's goal?

Line 424 "The finiteness and boundedness of V impose stringent requirements on
positional encoding, which, to some extent, necessitates unbounded positional encoding" It's unclear what this statement means.

---

> ### Author Response · Authors · 2024-11-24
> **Author's responses**
>
> Thank you for your careful review and thoughtful comments. Below, we provide our responses:
>
> - **Structure and Writing**
>   Thank you for your remarks. We have thoroughly reviewed the manuscript and addressed the identified notational and grammatical issues. Additionally, we have revised the proof of Theorem 8 to improve its readability. While we acknowledge that the notations for network structures occupy significant space, they are indispensable for presenting our results with the necessary clarity and rigor.
>
> - **In-context Learning in Lemma 3 and Theorem 8**
>   Thank you for your insightful questions. The primary objective of this paper is to theoretically analyze the relationship between the Transformer architecture, in-context learning, and positional encoding.
>
>   Lemma 3 examines in-context learning in an unconstrained setting, as you noted. Although this setup diverges from practical scenarios, it aligns with prior theoretical studies (e.g., Petrov et al., 2024b) that explore universal approximation properties. Since practical in-context learning is highly complex and challenging to analyze directly, this provides an important foundational step.
>
>   In Theorem 8, we shift focus to in-context learning under the constraint of a finite vocabulary, providing a stark contrast to Theorem 6. The results demonstrate the substantial differences between employing positional encoding and omitting it. This distinction underscores the benefits of positional encoding, which is a central theme of our manuscript.
>
>   Your observations also suggest intriguing directions for future exploration, such as comparing the function approximation capabilities of the approach in Lemma 3 with direct utilization of FNNs, particularly when the Transformer parameters are trainable. We have added a brief discussion of these points to the *Discussion* section as potential avenues for future work.
>
> - **Line 398: Previous work.**
>   This refers to the conclusions presented in Section 2 of the manuscript. We have revised the text to clarify the intended meaning and avoid ambiguity.
>
> - **Line 383: Typo.**
>   Thank you for reading carefully. Here, we missed specifying $P_y = 0$ in this context. The statement of the theorem has been updated accordingly in the revised manuscript.
>
> - **Line 409: Finiteness.**
>   The sets $\mathcal{V}_x$ and $\mathcal{V}_y$ are defined in the context of practical applications, and their elements are finite in number. Thus, the term *finiteness* indicates that the column vectors of $ UY $ and $ X^T B^T C $ can only take on a finite number of distinct values. We have revised the text.
>
> - **Line 410: Density of $S$.**
>   Thank you for raising this thought-provoking question. In our work, we consider Transformers with input sequences of arbitrary length, implying that the positional encoding $ P_x $ consists of a countably infinite set of elements (independent of the target function $f$). Consequently, $ S $ is also infinitely large and may or may not be dense in $ \mathbb{R}^d $. In Theorem 8, we assumed a strong density condition, which is later relaxed in Corollary 9. Additionally, based on another reviewer’s suggestion, we have further weakened the conditions for the exponential Transformer case (see Theorem 9 in the revised manuscript).
>
>   In practical applications, input sequences are not arbitrarily long but are always truncated. This allows our conclusions to be interpreted through an approximation lens, where the objective is to approximate functions within a specified error margin rather than requiring infinitesimal precision.
>
> - **Line 424: Boundedness of $V_x$.**
>   The intent here is to indicate that $ V_x $ contains only finitely many elements, rendering it bounded. For $ S = V_x + P_x $ to be dense in the entire space, $ P_x $ must be unbounded. In Corollary 9, we relaxed this requirement, eliminating the need for $ P_x $ to be unbounded, making the conditions more aligned with practical scenarios. We have clarified this point in the revised text.
>
> - **No experiments.**
>   Thank you for your understanding regarding the theoretical nature of this work. While we were unable to include experimental results, we hope that our findings provide valuable insights for experimental researchers. To this end, we have added a discussion in the *Conclusion* section, encouraging future studies to investigate the density properties of positional encodings experimentally.
>
> We sincerely appreciate your valuable feedback and hope that these clarifications address your concerns.

---

> > ### Comment · Reviewer_Tz2M · 2024-11-29
> >
> > Thank you for the response. However, my main concern about the premise of the paper (that one would optimize over the context to find the correct response during in-context learning) still remains. I don't find the argument that this setting aligns with past theoretical studies to be sufficiently convincing to justify why such a result is interesting.
> >
> > I retain my score of a 5, but will note to the area chairs my low level of confidence as I am unable to check the proofs for mathematical soundness.

---

### Official Review · Reviewer_dzSm · 2024-11-04

**Soundness:** 2
**Presentation:** 3
**Contribution:** 3
**Rating:** 5
**Confidence:** 4

**Summary:**

The paper presents several theoretical results on the expressivity of
individual transformer models with fixed weights implementing different
functions by changing their context.

* The paper considers transformers that make predictions given a sequence of
  embedded $x$/$y$ pairs as tokens in context.
* The transformer architecture has a single-layer, single-head, attention-only
  (no MLP) transformer (that is, essentially just a single attention
  mechanism). In detail, the architecture computes query--key affiliation
  scores based on the $x$-components of the tokens, processes these using
  either a softmax transformation or element-wise ReLU activation, followed
  by multiplication by a value score based on both $x$s and $y$s, after which
  they extract the prediction corresponding to the final output.
* The paper considers several settings and whether or not a transformer with
  arbitrary fixed (full-rank) attention matrices achieve the universal
  approximation property, in the sense that for any continuous function on a
  compact domain there exists a context of some length that causes the
  transformer's prediction of the next $y$ as a function of the next $x$ is
  arbitrarily close to the continuous function.
  1. If any real vectors are allowed as $x$/$y$ tokens in the context, the
     paper shows that the transformers have the universal approximation
     property in the above sense.
     * The proof relies on constructing a context of length $n$ that makes
       the transformer implement a given MLP with a single hidden layer of
       width $n$, and the classical result that such MLPs have the universal
       approximation property.
  2. If the $x$/$y$ tokens that are allowed are restricted to a finite set of
     pairs, then the paper argues that transformers lack the universal
     approximation property.
     * The argument proceeds by constructing a similarly constrained set of
       MLPs that are constructed using a finite collection of hidden units,
       and showing that these families of functions lack the universal
       approximation property, then arguing that the connection between MLPs
       and transformers from the previous setting shows that this is also the
       case for transformers.
  3. If the $x$/$y$ tokens are restricted to a finite set containing at least
     certain irrational basis vectors, *and* the transformer's inputs are
     augmented with an additive positional encoding that, when added to the
     $x$-component of the finite tokens, creates a set of tokens that are
     dense in the input space (or, in the case of ReLU networks, dense in at
     least the unit hypercube), then the transformers again achieve the
     universal approximation property.
     * I have tried to understand the statement of this theorem but given my
       expertise is not in approximation theory, I was unable to review this
       proof in detail. A superficial summary is that the proof involves an
       application of Kronecker's Theorem on approximating real vectors with
       integer multiples of irrational vectors.
* The paper motivates the importance of the above results with the
  observation that transformers used in natural language processing involve a
  finite vocabulary, which gives rise to a finite set of embedded token
  vectors. Therefore, an informative analysis of the in-context expressivity
  of transformers should involve such a finiteness constraint. In this
  context, the paper's results demonstrate that the inclusion of a positional
  embedding is crucial to retaining the universal approximation property.

I also include a summary of my review as follows.

* While functional approximation is not my area of expertise, I found the
  paper interesting and thought-provoking, and relatively clearly written and
  relatively easy to follow. I particularly liked the framework for studying
  transformers implementing different functions in context, and the neat
  construction of how to implement an arbitrary MLP using the attention
  mechanism.
* Unfortunately, I found what appears to be a potentially serious gap in the
  argument that the finite-vocabulary transformer architecture does not have
  the universal approximation property, and I was not able to see an easy way
  to close this gap. If I am correct and the authors are unable to close this
  gap then this would appear to undermine one of the major results of the
  paper.
* Aside from this, my overall impression of the novelty and significance of
  the paper was weakened by the strength of some of the assumptions and a
  lack of detailed comparison to what appears to be a closely related work
  (the cited work of Petrov et al., 2024b).
* While studying the paper I also noted a number of what seem to me to be
  minor technical errors that would probably be easy to fix, I list these
  along with questions to the authors I encountered while studying the paper.

**EDIT TO ADD:** Summary of discussion:

* The authors' revision fixed the gap I had noted in my initial review, but when looking at the paper in more detail during the discussion period I noticed another major gap in the proofs for the non-universality of finite-vocabulary prompting for softmax attention (see comment ['Response to Rebuttal Part 1'](https://openreview.net/forum?id=YE6N8htoFQ&noteId=fYd5DD26co)). Unfortunately due to this remaining gap I am still recommending that the paper should be rejected.
* The authors' revisions and rebuttals somewhat addressed the other concerns I listed, by clarifying the relationship with prior work and the role of the various assumptions.
* The authors' revisions addressed all of the minor technical errors and the authors answered all of my questions from my initial review. During the discussion period I noticed some additional minor technical issues in the revised paper which I have communicated to the authors.

**EDIT TO ADD:** Summary of further discussion and score updates.

* The authors outlined how to address the new gap I noticed. I am satisfied that their proposed fix will work. I am aware of no further gaps in the proofs for sections 2 and 3. I haven't been able to verify the proofs in section 4.
* The revised paper made explicit that section 4 does not apply to softmax attention, which I think is a significant limitation.
* The revised paper has improved the presentation, but various non-trivial presentation issues remain and I believe that resolving them requires further review, which is now impossible.
* Overall, while I am no longer aware of any major flaws in the argument, I still don't think the paper should be accepted. But I am less sure about this, so I am raising my score from 3 (reject) to 5 (borderline reject).

**Strengths:**

I think understanding the expressivity of neural architectures is an
interesting and important theoretical problem in deep learning. It is
important to have a clear understanding of the theoretical limits of our
models, and, while in my opinion there is often a disconnect between positive
expressivity results and the way that neural networks learn to implement
functions in practice, we can still derive qualitative insights about how
neural networks might implement certain kinds of functions using features
such as depth, or, in this case, an attention mechanism, which can be
informative in practice.

Within this topic, the current paper presents an analysis of the problem not
of the expressivity of transformers as a neural architecture, but of the
expressivity of an arbitrary transformer model with fixed weights through
changes to the prompt alone. This is an ambitious undertaking and has the
potential to shed light on one of the most important topics in modern deep
learning, namely the nature of in-context learning.

In this setting the authors have put forward an elegant notion of the
in-context expressivity of a fixed transformer through the provision of a
particular context. While as I have mentioned functional approximation is not
my area of expertise, it appears to me that this framework is novel and I
believe it has been well done.

A neat example of the in-context framework is the link the authors have
achieved between single-hidden-layer neural networks and their in-context
transformer (captured in Lemma 2 for the case of ReLU attention and embedded
in the proof of Lemma 3, though I have not reviewed the latter). I found this
connection interesting and thought-provoking, and it leads to a very elegant
proof of the universal approximation property of prompting a fixed
transformer in the setting with arbitrary token vectors.

**Weaknesses:**

**Gap in Theorem 6:**
Theorem 6 is accompanied by a very brief proof that says the result
immediately follows from the connection between FNNs and transformers plus
Lemma 5. I couldn't see how this conclusion follows from these results, and
in fact I have come to suspect that it might not follow from them at all.

Consider the case of ReLU networks. By "the connection between FNNs and
Transformers" I take it you are referring to Lemma 2. As far as I can tell,
the Lemma is one-directional, showing the existence of a context for every
FNN but *not* the existence of an FNN for every context. At the very least,
the reverse direction would require further justification.

Given this, in Theorem 6 it is not sound to reason that because there are
some functions that cannot be approximated as an FNN with ReLU activation
these same functions must not be approximable by the transformer with any
context. It seems to me that you also have to rule out the existence of
another context which might approximate the function.

In the case of softmax networks the same issue may apply, however the status
is not clear to me because there is no separate Lemma for "the connection
between FNNs and Transformers" for the case of softmax activation, with the
details apparently to be found in the proof of Lemma 3 and involving
reasoning via a custom architecture involving exponential activation.

I don't immediately see how this argument can be recovered, and I invite
clarification from the authors. If I am not mistaken about this problem with
the proof of Theorem 6, and it is not able to to be resolved, then I can't
recommend the paper for acceptance.


**Inadequate discussion of closely related work:**
The authors state in the introduction "Meanwhile, Petrov et al. (2024b)
explored the role of prompting in Transformers, proving that prompting a
pre-trained Transformer can act as a universal functional approximator." I
was not previously familiar with this cited work, but from the authors' own
description it sounds quite closely related in scope to the present work,
which is also about proving that under certain conditions a transformer can
act as a universal function approximator.

Could the authors please clarify the relationship between the contributions
of Petrov et al. (2024b) and their contributions?

**Some implausible assumptions:**
I believe the authors are interested in finding universal approximation
results that will eventually speak to the limitations of architectures
used in practical deep learning settings. Given this motivation, I was made
uncomfortable by the following features of the setting and assumptions.

1. **The use of ReLU attention.** Of course, ReLU is a very commonly used
   activation function, including with transformer architectures. However, I
   have never seen it used in place of softmax for the post-processing of the
   query--key affiliation scores (usually it would be used, for example, as
   part of an MLP step after the attention step in a transformer block).

2. **The reliance on a dense positional encoding.** In Theorem 8 the
   universal approximation property is achieved under the assumption that the
   positional encoding essentially turns the finite input token vocabulary
   into a set that is dense in $\mathbb{R}^{d_x}$.

These features are apparently in a kind of trade-off: In Corollary 10, the
authors give a universal approximation result requiring the positional
encodings are merely dense in the unit hypercube, not the whole input space.
This is also a tall order but seems much more plausible. However, this
Corollary is only given for transformers with ReLU attention, not the more
standard softmax attention.

Finally, I appreciate that the authors have acknowledged the strong
assumption that the positional encoding is dense, and pointed out that they
can be addressed with future work. However, I would have liked to see a more
in-depth discussion around this topic: do the authors have any reasons to
believe that this assumption could be relaxed, or does it appear to be
fundamental to the entire diophantine approximation approach pursued here?

**Questions:**

I studied the technical results up to and including the statement of Theorem
8, and I noticed the following potential minor errors, all of which I expect
would be easy to fix (if I am not mistaken about them in the first place). I
would be happy to clarify any of my questions in further detail as needed.

Definition of transformer and feed-forward architectures:

1. In the definition of attention, $Q$ and $K$ have undefined shapes (only the
   shape of $B$ and $C$ are defined). One can infer from usage that $Q$ and
   $K$ have $d_x + d_y$ columns, but the number of rows could be any number
   greater than or equal to $d_x$ and the equations could come out the same.
   I invite the authors to consider removing the zero rows entirely such that
   $Q = [B\ 0]$ and $K = [C\ 0]$.

2. In the definition of attention, $V$ is described with shape $d_y$ by $d_y$.
   This must be a mistake since:
   * $V$ is multiplied by $Z$ of shape $d_x+d_y$ by $n+1$ implying it should
     have $d_x+d_y$ columns.
   * The shape of the output of attention has the same number of rows as $V$,
     and needs to be added to $Z$ (equation 10), suggesting that $V$ should
     have $d_x+d_y$ rows too.
   * Indeed later (line 266) the authors partition $V$ into blocks such that
     the shape is $d_x+d_y$ by $d_x+d_y$.

3. In Equation 10 there is an undefined symbol $h$ which, from context,
   appears should be the activation function $\sigma$.

4. Equation 10 uses input $x; Z_{:, 1:n}$ whereas later invocations of
   $T^\sigma$ use $x; X, Y$ (and the RHS of equation 10 is expressed in terms
   of $Z$).

5. I invite the authors to consider promoting some assumptions on the
   transformer architecture from later in the text to section 2.1 where the
   architecture is introduced, so that they are all in one place.
   * This applies to the decomposition of $V$ into four parts.
   * Also the assumption that $B$, $C$, and $F$ are non-singular would then
     make sense in section 2.1.

6. As written, the definition of feed-forward networks does not appear to
   allow for the use of softmax activation, which is not an element-wise
   function due to normalisation.

    This led me into some confusion later in the paper when the authors talk
    about how softmax FNNs with a finite vocabulary of units leads to an
    infinite-dimensional family of functions. Could the authors please clarify
    the definition of FNNs with softmax activation and the definition of the
    finite-vocabulary family of softmax networks, if they indeed intend for
    these networks to have normalisation?

7. Finally, in equation (11) (the definition of the finite-vocabulary family
   of transformers), $n$ is fixed, but I think the authors intended for it to
   be any positive integer. This would make more sense by analogy to
   classical FNN approximation results for unbounded width (see also the
   correspondence between FNN width and context length of Lemma 2), and
   in Theorem 8 the authors explicitly allow unbounded context length.

   It seems important that the definition of the family should allow
   unbounded context length, because the authors want to say, for example in
   Theorem 8, that "[the family] can achieve the UAP", and in Theorem 6, that
   it cannot, but, trivially, if the family uses a finite context length then
   (given the tokens are also finite) it is a finite family of functions and
   therefore it trivially cannot have the UAP.

Lack of universal approximation properties for finite vocabulary setting:

8. What norms are being used in the approximation property statements?
   Starting with Lemma 1, which I think is the norm used throughout, but then
   also for Theorem 6, there is a norm with subscript $C(K)$ (is that the
   same?)

9. What do the authors mean on line 308 by "the case of non-softmax
   activation"? I think they simply mean ReLU activation, but I am left
   uncertain as to whether they are trying to make a more general claim.

10. What do the authors mean on line 309 by "It is well known that
   finite-dimensional spaces are compact"? This is false in the generality
   stated. Am I missing an assumption? It seems to me that the span would be
   unbounded and therefore it is not compact. Nor does the family of networks
   appear to be a closed and bounded subset of the span, which would ensure
   compactness given that the span is finite-dimensional.
   Actually, I am not immediately sure how to resolve this, but I hope the
   authors might be able to address it.

    As an aside, I invite the authors to consider stating the ReLU case as a
    numbered lemma and giving a formal proof, even if turns out to be a short
    proof.

11. On line 312 the authors state that "the dimension of the span of [the
   finite set of softmax networks] might be infinite". I didn't immediately
   understand this claim, and I wanted to check my understanding. Is is due
   to the presence of normalisation between units that even though the softmax
   networks are each comprised of weighted units from a fixed finite
   collection of basic units, the normalisation means that these networks
   won't generally be linear combinations of each other?

12. In Lemma 5, the statement must hold for any $\epsilon_0 > 0$. Intuitively
   I thought the choice of inapproximable function should have to depend on
   $\epsilon_0$, but I don't see such a dependence in the proof sketch, and
   in the proof it is stated that $\epsilon_0 = 0.1$. I am concerned that the
   proof does not go through for $\sin(m \pi x)$ if $\epsilon_0$ is large
   enough since the approximation will no longer have to distinguish between
   zeros and peaks.

    Perhaps the constructed function should be something more like
    $\frac{10}{\epsilon_0}\sin(m \pi x)$? (I'm not sure what norm is being
    used but I chose $10$ based on the decision to use $\epsilon_0 = 0.1$ in
    the proof).

Restoration of universal approximation property through positional encoding:

13. Theorem 8: In the statement, there is no introduction of the function $f$,
   which I assume should be introduced as a continuous function from the
   compact domain to $\mathbb{R}^{d_y}$.
14. Line 383: In the definition of $S$, you want $j \in \mathbb{N}^+$ rather
    than $i \in \mathbb{N}^{+}$.
15. Line 431: The same error again, you want $j \in \mathbb{N}^+$.

Typos:

16. Line 58: "Transformersin" missing space.
17. Line 111: "for for".
18. Line 283: "Propriety" in title of section 2.4.
19. Line 383: "and ,"

---

> ### Author Response · Authors · 2024-11-24
> **Validity of Theorems**
>
> We sincerely thank you for your careful review, thoughtful comments, and valuable suggestions. Your detailed feedback has greatly helped us improve our manuscript. Below, we provide our responses in two parts: the first addressing the validity of the theorems, and the second responding to additional comments and questions.
>
> ### **Part 1: Validity of Theorems**
>
> - **Gap in Theorem 6**
>
>   Thank you for pointing this out. Upon reviewing Theorem 6 in light of your comments, we identified gaps in the original proof. We have now carefully revisited the entire manuscript, reorganized the relationship between lemmas and theorems, and corrected the proof.
>
>   The proof of Theorem 6 requires separate considerations for the ReLU and softmax cases:
>
>   1. **ReLU Case**:
>
>      Lemma 2 establishes an equivalence relationship. Specifically, any single-layer Transformer with in-context learning can be rewritten as a single-hidden-layer feedforward neural network (FNN). We have emphasized this in the revised version. Conversely, every single-hidden-layer FNN can also be represented by a single-layer Transformer with in-context learning, as highlighted in the original Lemma 2. This equivalence, combined with the limited expressive power of $ \mathcal{N}_*^{\text{ReLU}} $ (now explicitly incorporated into Lemma 5 in the revised version), leads directly to the conclusion of Theorem 6.
>
>   2. **Softmax Case**:
>
>      Unlike the ReLU case, the equivalence relationship in Lemma 2 does not hold here, creating a gap in the original proof. In Lemma 5, we demonstrate that $\mathcal{N}_*^{\text{softmax}} $ also has limited expressive power. While this lemma alone does not directly establish Theorem 6, its proof techniques can be extended to achieve the desired result. The key distinction between
>
>      $\mathcal{N}_*^{\text{softmax}} $
>
>      and $ \mathcal{T}_*^{\text{softmax}} $ lies in the additional normalization component used. The complete proof can now be found in Appendix B.
>
> - **Related Work**
>
>   We have added a discussion on Petrov et al. (2024b), who explored universal approximation properties (UAP) in the context of in-context learning, but without considering vocabulary constraints or positional encoding. Their results align with our Lemma 3, and this connection has been noted in the manuscript.
>
> - **Implausible Assumptions**
>
>   * **The use of ReLU attention:** We chose the ReLU activation function primarily for its simplicity. The revised manuscript now addresses the case of general elementwise activations. Proving the case for softmax activation requires more intricate techniques, and we have added a discussion on this point.
>
>   * **The reliance on a dense positional encoding:** The current proof directly relies on the UAP of FNNs without weight constraints, but we believe the density condition can be further refined. In practice, FNNs can achieve UAP even with constrained weights, as demonstrated by ReLU FNNs. Additionally, for the exponential activation case, we have relaxed the conditions, further emphasizing the versatility of our approach (see Theorem 9 in the revised manuscript).
>
>   * **Corollary 9:** We have replaced Corollary 9 with Theorem 9, adding conclusions for the case of exponential activation functions under slightly weaker conditions than in the ReLU case. Specifically:
>     - For the ReLU case, leveraging its positive homogeneity, the density condition on $ S $ can be relaxed to density within a cube containing the origin.
>     - For the exponential case, we utilized properties of higher-order derivatives of exponential functions, which relate to polynomials. By approximating higher-order derivatives with finite differences, we achieved the desired result.
>
> We hope these revisions address your concerns and improve the overall quality of the manuscript. Thank you again for your invaluable feedback.

---

> ### Author Response · Authors · 2024-11-24
> **Responses to Questions**
>
> ### **Part 2: Responses to Questions**
>
> - **Questions 1–4:** Thank you for your detailed review and suggestions regarding the notations. We have revised the manuscript to correct typos and enhance readability.
>
> - **Question 5: Assumptions:** We have added an "Assumption" to consolidate the major assumptions in one section for clarity.
>
> - **Question 6: Softmax FNNs:** Following your suggestion, we have introduced the notation for softmax FNNs in the revised manuscript.
>
> - **Question 7: Unbounded context length $ n $:** The context length $ n $ is unbounded throughout the manuscript. The distinction between Theorem 6 and Theorem 8 lies in the use of positional encoding. We have revised the equations and accompanying text to clarify this point.
>
> - **Question 8: Norms involved:** The UAP considered pertains to uniform norm approximation. The norm statements in Lemma 1 and Theorem 6 are equivalent, and we have updated the notations for greater clarity.
>
> - **Question 9: Non-softmax activation:** Here, we specifically refer to ReLU activation. Since many results hold for general elementwise activations, we have revised the statements of the lemmas and theorems accordingly.
>
> - **Question 10: Finite-dimensional spaces:** In functional analysis, "finite-dimensional spaces are compact" refers to compactness under the functional norm, which differs from compactness in terms of function values. We have revised the wording to avoid ambiguity and incorporated this clarification into Lemma 5.
>
> - **Question 11: Dimension of span (softmax NN):** You are correct that this is due to normalization. We have clarified this in the revised manuscript.
>
> - **Question 12: $ \varepsilon_0 $ in Lemma 5:** You are correct; while the statement was accurate, it did not align perfectly with the proof, which pertains to the existence of $ \varepsilon_0 > 0 $. We have simplified the statement accordingly.
>
> - **Question 13: Function $ f $ in Theorem 8:** The function $ f $ is a continuous mapping from a compact domain to $ \mathbb{R}^{d_y} $. We have updated the statement for clarity.
>
> - **Questions 14–19:** Thank you for your meticulous reading. We have reviewed the manuscript thoroughly to address grammar issues and typos.

---

> > ### Comment · Reviewer_dzSm · 2024-11-27
> > **Response to Rebuttal Part 2**
> >
> > Thank you for responding to each of my questions in detail. I checked these revisions carefully and I am totally satisfied with most of the answers and the accompanying revisions (I think they really did significantly enhance readability). I have a few follow-up questions and notes.
> >
> > (To be clear: This is regarding part 2 of your rebuttal. Regarding Part 1, on validity of theorems, I want to work through the revised paper to check the new theorems, and will respond ASAP about this hopefully within a day or so.)
> >
> > **Q2:** Looks great! I noticed now in the decomposition of $V$ you assume the top-right block $E$ and bottom-left block $F$ are zero.
> >
> > 1. Is this because the theorems didn't hold given the originally-stated sparsity pattern for the value matrix? My guess is, in particular, the converse direction of the correspondence between NNs and transformers didn't hold? (I am yet to work through the revised statements and proofs in detail).
> > 2. I had a quick look at the assumptions and justifications in prior work. I saw that Cheng et al. (2024) considers essentially the same sparsity pattern for $V$ (though in their case $d_y=1$), and they also cite Anh et al. (2024), but I couldn't see this assumption in Anh et al. (2024). Could you please point it out?
> > 3. Accordingly I couldn't find a justification for this sparse partition of $V$. Could you comment on what this assumption means in terms of any limitations it places on the transformer architecture's computation? Why do you think it's a reasonable assumption?
> > 4. Limiting the architecture like this does not undermine the result that certain transformer families have the UAP, but it does raise the question of whether transformers with a finite vocabulary and no such sparsity restrictions would still *lack* the UAP. I don't expect you to have a precise answer on this, but do you agree that this is a potentially important assumption for the negative result, and do you have any intuition about whether sparsity is going to affect this?
> >
> > **Q7:** Looks great! Two minor formatting suggestions:
> >
> > 1. Throughout for your large set-builder expressions you might like to [use `\middle|` to grow the central vertical bar symbol to the same height as the `\left\{` and `\right\}` braces](https://tex.stackexchange.com/a/456).
> > 2. There is a stray space at the beginning of line 229 (possibly due to an un-escaped newline in the command for colouring the text?)
> >
> > **Q8:** Thank you for clarifying! Now I understand all of the statements clearly. Notes:
> >
> > 1. Lemma 5 is still written with the $||-||_{C(\mathcal{K})}$ notation. It might make sense to change to the same formulation via $\max$ as in Theorem 6.
> > 2. At that point if I am not mistaken all norms given in statements in the main text would be uniform norms of vectors, and there would be no function space norms.
> > 3. I quickly skimmed the appendix and the only other norm notation I spotted was the vector norm notation in the statement and proof of Lemma 12: $||-||_{\max}$. Is this also uniform norm?
> > 4. The notation you use is up to you, and at this point I think the paper is acceptably readable, but for what it is worth when considering vectors, the uniform norm notation I am most familiar with is to explicitly indicate a subscript $\infty$ as in $||-||_{\infty}$.
> >
> > **Q10:** Thanks for the clarification! I don't have a background in functional analysis, so please accept my apologies for my further novice questions, but I am committed to understanding this part of the paper.
> >
> > 1. I noticed you now write $\mathcal{N}^\sigma_\ast \equiv \mathrm{span}(\mathcal{N}^\sigma_\ast)$ (line 336.5). Do you mean $\mathcal{N}^\sigma_\ast \subseteq \mathrm{span}(\mathcal{N}^\sigma_\ast)$? It seems that the architecture allows only linear combinations of vocabulary elements (units) with non-negative integer coefficients, which is going to be countable. In contrast the span would include any real-coefficient linear combinations of units which is an uncountable set of functions. So they can't be equal. (Or, I am missing something).
> > 2. I just don't see how in full generality a finite dimensional space of functions is compact under even the uniform (function) norm. Here is an apparent counterexample: Consider the set of element-wise ReLU neural networks with weight vocabulary $\lbrace (1, 1, 0) \rbrace$, as in, the finite-unit family of networks is $\lbrace 0, \mathrm{relu}(\cdot), 2 \mathrm{relu}(\cdot), \ldots \rbrace$. Say the compact domain is $[0, 1]$. Then the elements in this family of functions still have unbounded function norm ($||0||^{C([0,1])} = 0$, $||\mathrm{relu}(\cdot)||^{C([0,1])} = 1$, $||2\mathrm{relu}(\cdot)||^{C([0,1])} = 2$, ...). I must be missing something, could you please refute this example?
> >
> > The rest all looks great, thank you again for your revisions. I am excited to review the remaining revisions in detail.

---

> > > ### Author Response · Authors · 2024-11-28
> > >
> > > Thank you once again for your professional review and valuable suggestions, which have significantly improved the quality of our manuscript.
> > >
> > > **Q2: The sparsity of $V$.**  We greatly appreciate your question and insightful comments. The main messages of our manuscript are:
> > >
> > > (1). All single-layer Transformers (without positional encoding) cannot achieve the UAP by VICL.
> > > (2). In contrast, there exist certain single-layer Transformers with appropriate positional encoding that can achieve the UAP by VICL.
> > >
> > > The assumptions on page 5 were introduced to facilitate the constructive proof of point (2). Specifically, assuming $F = 0$ simplifies the construction.
> > >
> > > We will revise the manuscript to incorporate the following clarifications:
> > >
> > > - **Regarding point (1):** All single-layer Transformers (without positional encoding) cannot achieve the UAP by VICL. Here, “all” means the assumptions on page 5 can be neglected. This argument is verified by our proof, where these assumptions are not essential.
> > >
> > > - **Regarding point (2):** The assumptions on page 5 are intended to simplify the construction for the UAP. We acknowledge that the practical reasonableness of these assumptions is unclear and leave this question for future investigation.
> > >
> > > - Except for Theorems 8 and 9, all other results are valid for general $E$ and $F$.
> > >
> > > - In Theorems 8 and 9, we assume $F = 0$ to simplify the construction (the case of $E = 0$ or $E \neq 0$ does not affect this argument). For $F \neq 0$, our current proof is not applicable and would require additional techniques, possibly with stricter assumptions on the positional encodings. Since the main goal of our work is to establish the existence of Transformers for VICL, we have not pursued these improvements. This remains an open problem for interested readers.
> > >
> > > Additionally, we will revisit the assumptions in Cheng et al. (2024) and Anh et al. (2024) to ensure our citations are accurate.
> > >
> > > **Q7.**  Thank you for pointing this out. The stray space in line 229 was due to a misalignment in the text. We will revise the formatting accordingly.
> > >
> > > **Q8.**  Thank you for your careful review. We will revise the notation to use only vector norms and update Table 1 to clarify that the norm $||\cdot||$ refers to $||\cdot||_{\infty}$. The subscript “max” in Lemma 12 is redundant, and we will remove it.
> > >
> > > **Q10.**  Thank you for identifying this issue. The correct statement should be $\mathcal{N}^\sigma_\ast \subseteq \mathrm{span}(\mathcal{N}^\sigma_\ast)$. Equality, i.e., $\mathcal{N}^\sigma_\ast = \text{span}(\mathcal{N}^\sigma_\ast)$,
> > > only holds when the parameters $a_i$ are unconstrained. The approximation power of $\text{span}(\mathcal{N}^\sigma_*)$ is limited, and consequently, the approximation capacity of $\mathcal{N}^\sigma_*$ is also limited. We will revise the statement to correct this error.
> > >
> > > **The compactness of a function space.**  For finite-dimensional Euclidean spaces, compact sets are equivalent to closed and bounded sets. Similarly, finite-dimensional function spaces are homeomorphic and isomorphic to Euclidean spaces of the same dimension. Your comment has made us realize that our previous statement contained an elementary error. The correct statement should be: “The unit ball in a finite-dimensional space is compact,” rather than “finite-dimensional function spaces are compact.” In fact, our proof only relies on the property that finite-dimensional function spaces are closed. We will add a reference to clarify this point. Thank you again for your review, which has helped us avoid this oversight.

---

> > > > ### Comment · Reviewer_dzSm · 2024-11-28
> > > >
> > > > Thank you again, this completely addresses all of my follow-up questions and I have no further questions at the moment.

---

> ### Comment · Reviewer_dzSm · 2024-12-01
> **Response to Rebuttal Part 1**
>
> Thank you for your detailed response to my concerns and your efforts to revise the paper. As I said in my response to rebuttal part 2 (questions), I do think you have managed to improve the presentation of the results. Your revisions have partially addressed my three concerns. Unfortunately, when reviewing the updated proofs for sections 2 and 3, I found another major gap in theorem 6 in the softmax case, and I still think the proof is incorrect.
>
> **On gaps in proofs:**
>
> * (ReLU/element-wise case) Thanks for your clarification and revisions. I have carefully reviewed the proof of Lemma 2 and I now see how the construction in Lemma 2 generalises to element-wise activation functions, and how the construction of the context is bi-directional (in the sense that it can be inverted to construct an FNN for each context). So, I am now convinced that the main non-universality result holds in the elementwise case. However, I still think the presentation of Lemma 2 could be substantially improved. For example, the statement does not currently claim bi-directionality, so to understand the invocation of the Lemma in the proof of Theorem 6, one has to understand details of the proof of the Lemma.
>
> * (Softmax/normalised case) Thanks for your revisions. I have carefully reviewed Lemma 11 and the proof of Lemma 5 and Theorem 6 in the softmax case. I followed the argument closely and, apart from some minor issues I will document in a future comment, I think that each step is sound. However, on reflection I realised that there is **another major problem in the proofs for Lemma 5 and Lemma 6 in the softmax case.** The problem was in the original submission (apologies for not noting it in my original review). The problem is as follows.
>
>     The goal is to prove that there exists a function $f$ for which no softmax FNNs or softmax transformer prompts approximate $f$. However, the proof constructs a function $f$ given a fixed size $k$ (the number of units in the FNN / the number of tokens in the prompt). The proofs crucially rely on the fact that $k$ is fixed for the construction of $f : f(x) = \sin(m x)$ where $\lceil m/\pi \rceil > k - 1$. While the proof shows that any particular constructed $f$ at size $k$ is non-approximable by any FNN / prompt with size $k$, this does not rule out the case where other FNNs / prompts in the finite-vocabulary family approximate the given $f$. Unfortunately I don't see any simple fix for this proof. It seems that the entire approach relies on $k$ being fixed.
>
> **On related work:** Thank you for adding the brief comparison to Petrov et al. (2024b). These revisions addressed my concern.
>
> **On assumptions:**
>
> * ReLU attention: I am impressed by the generalization the broader class of element-wise activations. I still consider element-wise attention less interesting than softmax attention.
> * The reliance on a dense positional encoding: Thanks for the explanation. I think it still remains to be seen how far this diophantine approximation-based approach can be taken but if the other issues are sufficiently addressed, I think this can be left to future work.
> * Corollary 9 / Theorem 9: Sounds like a neat result, nice work.
>
> **Summary:** My current understanding of the soundness of the paper's theoretical results is as follows:
>
> * The results in section 2, on approximating FNNs with transformer prompts and therefore universality of transformer prompting, appear to hold for both element-wise and softmax architectures.
> * The results in section 3, on the non-universality of finite-vocabulary architectures, appear to hold for element-wise activation but **they have not been successfully proven in the case of softmax activation.**
> * I have not checked the soundness of the results in section 4.
>
> Based on this understanding, since one of the major claims of the paper (that a transformer with softmax attention with finite vocabulary but without positional encoding) has not been successfully established, I still think the paper should be rejected.

---

> ### Comment · Reviewer_dzSm · 2024-12-01
> **Additional notes on revised paper**
>
> While reviewing the revised paper, I noted the following additional minor issues.
>
> Definitions:
>
> 1. In the definition of FNN architectures (equations 5, 6, and 7), you have $k \in \mathbb{N}$, but for transformers, you consider prompts with lengths in $\mathbb{N}^{+}$. It seems like you want to use the same set in both cases because you want to set up a correspondence between the architectures.
> 2. Relatedly, there is no softmax FNN with zero units, due to the impossibility of normalisation. This requires a special case definition. Currently your equation (7) is not well-defined in the case $k=0$.
>
> Lemma 11:
>
> 3. Consider swapping the order of appendices A.1 and A.2 to mirror the order the results are discussed in the main text. Currently the proof of Lemma 11 comes after the proof of lemma 2.
> 4. In the proof of Lemma 11, the desired size of the softmax network is not made clear, and the notation $W_1$ is undefined.
>     * Note, it seems the proof can be completed with a softmax network with only $k$ units instead of $k+1$ units. When verifying the proof, this is how I made sense of the ambiguous notation ($W_1 = W$). However, I ran into problems later in the proof of Lemma 3, where you rely on the existence of this extra unit. I think it's important to be very clear about the shape of the construction since it is important for the later proof, yet not strictly necessary for this proof.
> 5. Line 868: you write $N^{\mathrm{exp}}$ but I think you meant $N^{\mathrm{exp}}(x)$.
>
> Lemma 2:
>
> 6. The statement and proof of Lemma 2 works for any element-wise activation function (and you invoke it as such later) but there are still multiple references to ReLU activation.
> 7. On line 813 you say that appendix A considers the case where $F \neq 0$, but as written, the proof of Lemma 2 requires $F=0$. As I think you are aware, you could fix this by defining $Y = U^{-1}(A-FX)$ instead of $Y = U^{-1}A$.
>
> Lemma 3:
>
> 8. Line 901: In the proof of Lemma 3 you cite and link to Lemma 3. I think you meant Lemma 11.
> 9. Line 899: Also, you say you are constructing an exponential FNN but I think you meant a softmax FNN.
> 10. Further to the ambiguity discussed in point (4) above, in equation (44) you are missing a $k+1$th row containing zeros.
> 11. In many cases on page 18, $\tilde x B^{\top} C \tilde x$ is meant to be $\tilde x^{\top} B^{\top} C \tilde x$.
>
> Lemma 5 (element-wise activation):
>
> 12. In the proof sketch after the statement and the proof in appendix B, you now correctly appeal to closedness rather than compactness, However, you write that $\mathcal{N}^\sigma_\ast$ is closed. This is not implied by the cited result (Rudin, 1991, Theorem 1.21), which only says that $\mathrm{span}(\mathcal{N}^\sigma_\ast)$ is closed. You should instead argue that $\mathrm{span}(\mathcal{N}^\sigma_\ast)$ is closed based on the cited results and that this implies that $\mathcal{N}^\sigma_\ast$ is not dense in the function space.
>
> Lemma 5 (softmax activation). As I noted, the proof fails due to the dependence of the constructed $f$ on $k$. Here are some more minor issues:
>
> 13. Strictly speaking, you should mention the case there $A_{j,i}\exp(b_i) = 0$ for all $i$ (i.e. $A_{j,i} = 0$ for all $i$), as this is a special case in your proposition.
> 14. Line 1028.5 missing spaces after "interval" and "either".
> 15. Line 1029: What is either $\lceil m/\pi \rceil + 1$ or $\lceil m/\pi \rceil$? The number of zeros in $f$ or in the hypothetical $N$? I think you may mean "at least" instead of "either" etc.?
>
> Theorem 6 (element-wise activation):
>
> 16. Line 1043: In the proof you say $T^\sigma = N^\sigma$. Perhaps you meant $\mathcal{T}^\sigma = \mathcal{N}^\sigma$? Otherwise, I think you are missing a subscript $\ast$?
> 17. I think there is another minor gap in this proof. It seems to me that you need to further establish that to the finite-vocabulary family of transformer functions there corresponds a finite-weight-set family of FNNs of the form you studied in Lemma 5. It is clear from the proof of Lemma 2 (though not the statement, as I noted in my previous OpenReview comment) that for each context there exists an FNN $N^\sigma \in \mathcal{N}^\sigma$. But you need to prove that the collection of all of these FNNs is a subset of some finite-vocabulary FNN family (with a finite weight sets $\mathcal{A} \times \mathcal{W} \times \mathcal{B}$). You have not made it clear how the vocabulary of the given transformer relates to the vocabulary of the FNN family on which you intend to invoke Lemma 5. I believe it probably exists, so the theorem is probably still correct in this case, but I think your proof ought to spell this out.

---

> > ### Author Response · Authors · 2024-12-01
> >
> > Thank you once again for your interest in our paper and for providing such detailed feedback and suggestions. Your response has been immensely beneficial to us.
> >
> > We carefully reviewed your comments and found them to be very reasonable. We will incorporate your suggestions into the revision to improve the readability of the paper, especially the appendix.
> >
> > Regarding the gap you pointed out between Lemma 5 and Theorem 6, the main issue lies in the definition of $ k $. Upon reviewing the latest version of the manuscript, we realized that the definition of $ k $ was not explicitly clarified. Here, $ k $ refers to the size of the vocabulary, not the width of the FNN or the length of the context. In the first version, we mentioned that $ k $ represents the size of the vocabulary, but this emphasis was inadvertently omitted during the revisions. We apologize for this oversight, which may have misled your judgment. We will revise the paper to make this point clearer.
> >
> > To illustrate this, consider the case of an FNN. The case for Transformers is similar. A softmax FNN with $ n $ neurons can be expressed as:
> >
> > $$N^{\text{softmax}}_{\ast}(x) $$
> >
> > $$= \frac{\sum_{i=1}^{n} a_i \exp(w_i \cdot x + b_i)}{\sum_{i=1}^n \exp(w_i \cdot x + b_i)},
> > $$
> >
> > where $ (a_i, w_i, b_i) $ are drawn from a finite set $ \mathcal{A} \times \mathcal{W} \times \mathcal{B} $, and the number of elements in this set is denoted by $ k $. By merging neurons with the same weights, the FNN can be re-expressed as:
> >
> > $$
> > N^{\text{softmax}}_\ast(x)
> > $$
> >
> > $$
> > = \frac{\sum_{i=1}^{k} \hat a_i \exp(\hat w_{i} \cdot x + \hat b_{i})}{\sum_{i=1}^{k} \exp(\hat w_{i} \cdot x + \hat b_{i})},
> > $$
> > where $ \hat{a}_i, \hat{w}_i, \hat{b}_i $ account for the changes due to merging. Although $ n $ can be arbitrarily chosen, the transformed form reduces to a fixed width $ k $, enabling us to use the construction $ f(x) = \sin(mx) $ in the manuscript to demonstrate that its expressive power is limited.
> >
> > We hope this explanation addresses your concerns.

---

> > > ### Comment · Reviewer_dzSm · 2024-12-01
> > > **Thanks, still worried about gap, will look more closely**
> > >
> > > Thank you for your prompt response.
> > >
> > > Regarding the gap, it is indeed possible that I was mistaken about the new gap in the proof. I will consider your response carefully and check the proofs of Lemma 5 and Theorem 6 once again tomorrow. Since the end of the discussion period draws near, here is my immediate response.
> > >
> > > 1. I understand the idea that you could in principle merge neurons with the same weights and turn an FNN of unbounded width $n$ into an FNN of fixed width $k$.
> > > 2. I don't immediately see how the same is true for the transformer, but I can see that it is plausible (I will consider more carefully tomorrow).
> > > 3. However, I recall seeing no indication anywhere that the paper that attempts to signal the difference between the weights $a_i$ and $\hat a_i$ etc.; if there are really networks before and after aggregation the same notation ($A$, $W$, $b$) is used for the weights in both cases.
> > > 4. I believe the example in your response is incorrect. To get the normalisation constant correct after aggregating units, you would also need to count the number of units with each $(\hat w_i, \hat b_i)$. (Again, I saw no accounting for this in any of the proofs I checked in detail.)

---

> > > > ### Author Response · Authors · 2024-12-02
> > > >
> > > > Thank you for your response.
> > > >
> > > > Previously, we considered this to be a straightforward fact and therefore did not elaborate on it in detail. The notations $ \hat{a}_i, \hat{w}_i, \hat{b}_i $ are introduced here to distinguish the coefficients. While these notations were not used in the original paper, we will introduce similar ones in the revision to make this fact clearer. The introduction of $ \hat{b}_i $ here already accounts for the counting factor, as it can be easily incorporated into the exponent. For instance, $ 10 \exp(w_i \cdot x + b_i) = \exp(w_i \cdot x + \hat{b}_i) $, where $ \hat{b}_i = b_i + \ln(10) $.
> > > >
> > > > In the case of a softmax Transformer, the only difference lies in the denominator. In fact, regardless of the specific form of the denominator, as long as it is strictly positive, the constructed function $ f(x) $ remains applicable. This is because the denominator does not affect the zero points, and our proof is fundamentally based on the properties of these zero points.
> > > >
> > > > Welcome any further questions or comments.

---

> > > > > ### Comment · Reviewer_dzSm · 2024-12-02
> > > > > **Final assessment**
> > > > >
> > > > > After spending a significant amount of time reviewing the revised theoretical results and in discussion with the reviewers, I have reconsidered my impression of the paper. Unfortunately, I still cannot recommend the paper for acceptance. However, I am raising my score from 3 (reject) to 5 (borderline reject), and will not fight against the paper's acceptance. A summary of my remaining concerns and my rationale for the new rating are as follows.
> > > > >
> > > > > * **I am not aware of any further gaps in the section 2 and 3 proofs.** I agree with the authors that with the FNN unit consolidation technique sketched in their responses above, the gap I flagged in Lemma 5 and Theorem 6 is not a fundamental problem.
> > > > >
> > > > >     I urge the authors to substantially improve their presentation of this proof technique. There are many places in the paper where $k$ stands for the number of hidden units. In my opinion the unit consolidation technique for finite-vocabulary networks is a central and non-obvious part of the argument that deserves explicit mention.
> > > > >
> > > > >     I haven't been able to check the remainder of the proofs of Lemma 5 and Theorem 6 in detail for minor issues, but I believe the overall proof technique works. I have not been able to check the proofs in section 4 in any detail.
> > > > >
> > > > > * **I still think the assumptions are too strong.** The revised section 4 and the revised discussion clarify that the main universality result with positional encoding results has been proven for elementwise activation only, and not softmax activation. This is a limitation I didn't notice during my initial review (I would claim it was not made transparent in the original submission except perhaps in the details of the proof of Theorem 8). This is a deal-breaker for me because, as I have said, I am mainly interested in softmax attention rather than elementwise attention.
> > > > >
> > > > >     I note that if theorem 9 is true, then I am no longer as concerned about the density assumption in particular (cf. my original review). Theorem 9 still requires a strong assumption on the positional encoding, but not as strong as Theorem 8's *density in an unbounded space.* In other words, while my original concerns were about both the density assumption and the activation assumptions, only my concern about activations remains.
> > > > >
> > > > > * **The paper needs further revision and review:** Having spent substantial time checking the revised proofs in detail, I can now step back and assess the presentation of the revised paper as a whole. The revised paper is improved compared to the original submission, and I appreciate the authors' efforts in this regard. However, there remain important non-trivial technical issues to address:
> > > > >
> > > > >   * First, there are the specific issues I have pointed out above since the latest revision. Some of these are trivial to fix, others I am not so sure about, such as issues (4) and (17) in my comment ["Additional notes on revised paper"](https://openreview.net/forum?id=YE6N8htoFQ&noteId=kxRGUmDWDy). There is also the matter of improving the presentation on the unit consolidation proof technique (as discussed above).
> > > > >
> > > > >   * Second, stepping back, I noticed that it is very hard to figure out which assumptions on the transformer architecture apply to which theorems. Stating the assumptions on the architecture upfront is helpful, but then these assumptions are casually waived at different times, and then resumed at others, and then there is even an appendix saying something about how the proofs work with different assumptions too. In my opinion the statement of each individual result should make it clear which assumption applies to that statement.
> > > > >
> > > > >   The authors have shown that they are committed to improving the presentation of the results. However, since the required revisions are non-trivial I believe they require further review (and unfortunately further revisions are impossible at this point both due to ICLR timeline and my capacity as a reviewer).
> > > > >
> > > > > **In summary,** while I am not aware of any gaps, I am unable to confidently certify the soundness of the main results, that the main theorems only cover the elementwise attention case and not the softmax case unfortunately detracts from their significance for me, and I think the paper could benefit from another round of revision and review before being ready for publication. Accordingly I am unable to recommend the paper for acceptance.
> > > > >
> > > > > However, since I the major gaps I had identified were addressed or mistaken, and I am less confident about where the bar should be set for sufficient presentation and significance of the lack of universality result for the positional encoding + finite vocabulary + softmax case, if the other reviewers or AC wanted to accept the paper I would not fight against them. Therefore I feel that my original rating of 3 (reject) is too low, and the appropriate rating that captures my final assessment is 5 (borderline reject).
> > > > >
> > > > > I thank the authors once again for their engagement during the discussion period.

---

> > > > > > ### Author Response · Authors · 2024-12-03
> > > > > > **The final acknowledgment**
> > > > > >
> > > > > > Thank you once again for your insightful comments and suggestions on our manuscript. As we mentioned previously, your feedback has significantly improved the quality of our work.
> > > > > >
> > > > > > Regarding the results in Section 4, we did not delve into the case of softmax functions. On one hand, this would require additional technical effort. On the other hand, we believe that Theorems 6 and 8, as presented, already convey the message we intended: as highlighted in the title and abstract, the presence or absence of positional encoding makes a significant difference for VICL. In the conclusion section, we briefly discuss the implications of this message for practitioners. While adding the softmax case could lead to stronger results, it does not provide additional insights beyond what is already conveyed.
> > > > > >
> > > > > > As for the presentation of our results, the original version indeed had several areas of concern, and we have revised the manuscript based on your and other reviewers' suggestions. The remaining typos (including those you pointed out, as well as other typos) have been corrected now. It is important to note that, for theoretical papers, it is nearly impossible to include every detail clearly. We must assume that readers have certain background knowledge to streamline the proof process, and the current version should be acceptable for most graduate-level readers in the sciences. On the other hand, being overly detailed might actually be detrimental, as it can overwhelm readers with unnecessary details, causing them to lose sight of the main points. Our current presentation strikes a balance between detail and readability.
> > > > > >
> > > > > > Regarding the assumptions in our paper, we have clarified that they are consistent throughout the manuscript. We only briefly mention more general cases after Theorem 6, referring readers to Appendix D for further elaboration. Including this appendix is logical and necessary for readers who wish to explore the message of the paper in greater depth. Here is why:
> > > > > >
> > > > > > - The key message of Theorem 6 is that Transformers without positional encoding, $T_\ast^\sigma$, do not possess the UAP. Theorem 8, on the other hand, shows that Transformers with positional encoding, $T_{\ast, \mathcal{P}}^\sigma$, can potentially have UAP. This difference is central to the paper. However, curious readers may wonder if the lack of UAP for $T_\ast^\sigma$ is due to the assumption on the $Q, K, V$ matrices in the Transformer. Appendix D clarifies that this assumption is not the cause.
> > > > > > - We did not include this detail in the main text to minimize the reader's cognitive load. Additionally, as mentioned by other reviewers, it is beneficial to group all assumptions together, which we fully agree with.
> > > > > >
> > > > > > Finally, regardless of whether the manuscript is ultimately accepted or not, we sincerely appreciate the careful and thoughtful effort you have put into reviewing our work.

---

> > ### Author Response · Authors · 2024-12-03
> >
> > Thank you for your suggestions. We have revised the paper according to your recommendations.
> >
> > * **Issue 1-2**: We have changed $k \in \mathbb{N}$ to $k \in \mathbb{N}^+$.
> > * **Issue 3**: Lemma 11 has been moved to the beginning, i.e., Appendix A.1.
> > * **Issue 4**: $W_1$ was a typo, and it should refer to $W$. In the proof, the width of the softmax FNNs is $k+1$, where $k$ is the width of the exponential FNN. We can indeed use a softmax FNN with $k$ neurons to complete the proof, but this does not affect the conclusion of the lemma.
> > * **Issue 5**: $N^{\exp}$ has been changed to $N^{\exp}(x)$.
> > * **Issue 6**: References to ReLU activation have been updated to $\sigma$ activations.
> > * **Issue 7**: The sentence "$F \neq 0$" at line 813 has been removed to make the statement in Appendix consistent with the main body of the manuscript.
> > * **Issue 8**: Line 901: "using Lemma 3" has been changed to "using Lemma 11".
> > * **Issue 9**: Line 899: "an exponential FNN" has been changed to "a softmax FNN".
> > * **Issue 10**: We only need to take $n = k+1$ here. After adding this relationship and fixing the typo involving $\tilde x$, Equation (44) no longer presents any issues.
> > * **Issue 11**: The typo $\tilde x B^\top C \tilde x$ has been corrected to $\tilde x^\top B^\top C \tilde x$.
> > * **Issue 12**: "$\mathcal{N}^{\sigma}_{*}$ is closed" has been changed to
> >
> > "$\text{span}(\mathcal{N}^{\sigma}_{*})$ is closed".
> > * **Issue 13**: The case where $A_{j,i}=0$ for all $i$ has been added to complete the proof.
> > * **Issue 14**: Spaces have been added.
> > * **Issue 15**: "the total number of zeros in the interval $[0,1]$ is either $ \lceil \frac{m}{\pi}\rceil+1$ or $\lceil\frac{m}{\pi}\rceil$" has been changed to "the total number of zeros of $(\mathrm{N}^{softmax}_{*}(x))^{(j)}$ in the interval $[0,1]$ is at least $\lceil\frac{m}{\pi}\rceil$."
> > * **Issue 16-17**: To accurately express the relationship between
> >
> > $\mathcal{T}^{\sigma}_\ast$
> >
> > and $\mathcal{N}^{\sigma }_\ast$,
> >
> > many notations would need to be introduced. However, a detailed characterization is not necessary for the proof of Theorem 6. It suffices to show that $\text{span}(\mathcal{T}^{\sigma}_{*})$ is also finite-dimensional. We have made the corresponding modifications:
> >
> >   The proof for the elementwise activation case in Line 1043 has been revised as follows:
> > "For cases of elementwise activations, since
> > $\mathrm{T}^{\operatorname{\sigma }}_{*}$ has a structure similar to
> >
> > $\mathrm{N}^{\operatorname{\sigma }}_{*}$,
> >
> > we find that $\text{span}(\mathcal{T}^{\sigma}_{*})$ is also a finite-dimensional function space. Hence, the same argument from Lemma 5 can be applied here to complete the proof."

---

> ### Comment · Reviewer_dzSm · 2024-12-02
> **Follow-up questions**
>
> Thanks for your clarification. In my haste, I was indeed mistaken about point (4) above. I now see what you mean about accounting for multiplicity using the bias term. I am going to revise the proofs again carefully now.
>
> I also want to review section 4 in some detail. I noticed some changes compared to the originally submitted version. Could you please answer the following questions:
>
> 1. Was statement of Theorem 8 in the original submission intended to apply also to softmax activation, or just ReLU?
> 2. The statement of Theorem 8 now explicitly includes only elementwise activations. Do any of the results in ~section 3~ (edit: I mean section 4) say anything about softmax activation?
>
> ---
>
> Edit: Never mind, I found the answer to my questions in the revised discussion:
>
> > the main results (Theorem 8 and Theorem 9) focus on elementwise activations. Future research should extend these findings to ... softmax activations. For softmax Transformers, our analysis in Sections 2 and 3 highlighted their connection to Transformers with exponential activations. However, extending this connection to the scenario in Section 4 proves challenging and requires more sophisticated techniques.

---

### Meta-Review · Area_Chair_7UwD · 2024-12-19

**Metareview:**

**Summary**: This paper investigates the universal approximation property (UAP) of transformers in the context of in-context learning (ICL), focusing specifically on the role of positional encoding. The authors provide theoretical results demonstrating that transformers with a finite vocabulary cannot achieve the UAP without positional encoding. However, they prove that the UAP can be achieved under certain conditions with the inclusion of positional encodings. The work bridges insights from approximation theory and neural architecture design, offering a formal framework for understanding the benefits of positional encodings in ICL tasks.

**Strengths**:
- Reviewers found the paper interesting and thought-provoking, filling a significant gap in understanding the theoretical limits of transformers in ICL tasks.
- Reviewers found this to be a unique perspective and fresh take on the expressivity of transformers.
- Reviewers found the theoretical insight and contribution to be compelling.

**Weaknesses**:
- Strong assumptions: Most reviewers had concerns that the strength of the assumptions limits and weakens the novelty and significance of the paper. My suggestion would be to include more discussion on how the results of this work could be generalized and applied more broadly to other PEs like RPE and RoPE, more discussion on practical insights for practitioners, and more discussion on where scenarios like "fixing the network weights and adjusting the content of the context so that the output of the Transformer network can approximate f" arise in practice. I don't agree with some of the reviewers that the paper needs experiments, but I do agree that it needs a discussion grounded in practicality to essentially tell the practitioner readers why should they care about this result.
- Opaque writing: Multiple reviewers complained about imprecise or opaque statements. In fact, Reviewer `dzSm` went through an extremely thorough and dedicated back and forth with the authors and after all this time they concluded that the writing is opaque, missing details or hard to follow. The reviewer spent a considerable amount of time trying to understand the proofs in detail, so the fact that it took them 3 rounds of back and forth to finally get clarity and believe the paper's claims tells me that the paper needs to go through a substantial writing refactoring and improve clarity/flow/ease of exposition before it can be accepted.

**Recommendation**: In my mind this is a borderline-ish paper, because the results seem exciting and everyone agreed it's a compelling finding, but at the same time the usefulness and importance didn't come through. Moreover, it appears that the theoretical weeds are still difficult to parse even for experts, and because of that I vote to Reject the paper and encourage the authors to go through another round of revisions.

**Additional Comments On Reviewer Discussion:**

See my discussion above, but essentially the very long, detailed, and ultimately productive back and forth between Reviewer `dzSm` and the authors convinces me that the paper has theoretical merit but is not quite there in terms of exposition. Additionally, everyone has concerns about the narrowness of the applicability of the result given the strong assumptions, so I think the authors need to put more effort into crafting a compelling "why should we care" motivation/generalization to more use cases.

---

### Decision · Program_Chairs · 2025-01-22

Reject